# Aneuploid embryonic stem cells drive teratoma metastasis

Rong Xiao[1,5], Deshu Xu[2,5], Meili Zhang[1,5], Zhanghua Chen ®[2,5], Li Cheng[1], Songjie Du[1], Mingfei Lu[1], Tonghai Zhou[1], Ruoyan Li ®[3,4], Fan Bai ®[2] ✉ & Yue Huang ®[1] ✉

Aneuploidy, a deviation of the chromosome number from euploidy, is one of the hallmarks of cancer. High levels of aneuploidy are generally correlated with metastasis and poor prognosis in cancer patients. However, the causality of aneuploidy in cancer metastasis remains to be explored. Here we demonstrate that teratomas derived from aneuploid murine embryonic stem cells (ESCs), but not from isogenic diploid ESCs, disseminated to multiple organs, for which no additional copy number variations were required. Notably, no cancer driver gene mutations were identified in any metastases. Aneuploid circulating teratoma cells were successfully isolated from peripheral blood and showed high capacities for migration and organ colonization. Single-cell RNA sequencing of aneuploid primary teratomas and metastases identified a unique cell population with high stemness that was absent in diploid ESCs-derived teratomas. Further investigation revealed that aneuploid cells displayed decreased proteasome activity and overactivated endoplasmic reticulum (ER) stress during differentiation, thereby restricting the degradation of proteins produced from extra chromosomes in the ESC state and causing differentiation deficiencies. Noticeably, both proteasome activator Oleuropein and ER stress inhibitor 4-PBA can effectively inhibit aneuploid teratoma metastasis.

Aneuploidy, defined as unbalanced numerical alterations either at the arm level or affecting whole chromosomes, has been recognized as a hallmark of cancer for more than 100 years[1]. Aneuploidy is a ubiquitous feature of cancer: approximately 90% of solid tumors and 70% of hematopoietic malignancies display a certain degree of aneuploidy[2,3]. Aneuploidy is usually caused by chromosomal instability (CIN), which elicits mitotic errors during chromosome segregation[4]. CIN is a driving force of tumor heterogeneity and evolution[5]; however, much less is known about the roles of aneuploidy in cancer progression. Systematic models of aneuploidy have hitherto been constructed by introducing extra chromosomes into euploid cells[6–9], thereby enabling the study of aneuploidy in the absence of CIN during carcinogenesis. Single-chromosome gains in yeast, mouse, and human cell lines have been found to impair proliferation and cause proteotoxic stress, replication stress, genome instability, and immune responses[10]. Concordant with the reduced proliferation potential of mouse embryonic fibroblasts (MEFs) and human colorectal cancer cells (HCT116) with particular types of trisomies, these cells are less tumorigenic[11]. While other studies show that trisomy 8 frequently arises during the culture of mouse embryonic stem cells (ESCs) and trisomies of human chromosomes 12,

[1]State Key Laboratory of Common Mechanism Research for Major Diseases, Department of Medical Genetics, Institute of Basic Medical Sciences, Chinese Academy of Medical Sciences, School of Basic Medicine, Peking Union Medical College, Beijing 100005, China. [2]Biomedical Pioneering Innovation Center (BIOPIC), Beijing Advanced Innovation Center for Genomics (ICG), School of Life Sciences, Peking University, Beijing 100871, China. [3]Wellcome Sanger Institute, Wellcome Genome Campus, Hinxton, Cambridge CB10 1SA, UK. [4]Present address: Department of Systems Biology, The University of Texas MD Anderson Cancer Center, Houston, TX, USA. [5]These authors contributed equally: Rong Xiao, Deshu Xu, Meili Zhang, Zhanghua Chen. ✉e-mail: fbai@pku.edu.cn; huangyue@pumc.edu.cn

17, or 20 always dominate the whole ESC population during prolonged cell culture, which endow human ESCs with the feature of neoplastic progression[6,12,13]. Thus, aneuploidy can exert both tumor-promoting and tumor-suppressive effects in a cell type-dependent manner[11,14,15]. However, in clinical observations, aneuploidy is often correlated with metastasis and poor prognosis[16–20]. The role of aneuploidy in cancer metastatic dissemination therefore merits further study. In this work, we use an aneuploid ESC-derived teratoma model to investigate the possible causal relationship between aneuploidy and cancer metastasis.

## Results

### Aneuploidy promotes ESC-derived teratoma metastasis

To assess potential metastasis of trisomic-ESC-derived teratomas, we performed subcutaneous injection of two single-trisomic mouse ESC lines with chromosome 11 or chromosome 15 gains (Ts11 and Ts15, where Ts represents trisomy)[6] and isogenic wild-type (WT) ESCs into severe combined immunodeficient (SCID) mice. About two to three months after teratoma assay, we occasionally observed lung metastatic lesions in some mice injected with Ts11 or Ts15 ESCs (Supplementary Fig. 1a, d). We confirmed that the metastases originated from the injected trisomic ESCs via the amplification of trisomy-specific fragments from the *piggyBac (PB)* transposon (Supplementary Fig. 1b, c). Histologic examination of the metastases (Met) showed the existence of undifferentiated or poorly differentiated cells (Supplementary Fig. 1d). In contrast, no metastasis was observed in mice bearing WT ESC-derived teratomas (Supplementary Fig. 1e). However, whether trisomy directly increases the metastatic potential of teratomas merits further exploration.

To systematically track metastatic colonization in vivo, we generated EGFP/Luc reporter ESC lines via the transduction of diploid and trisomic ESCs with the *EF1α-EGFP-Luciferase* lentivirus (Fig. 1a). One diploid (WT), four single-trisomic (Ts6, Ts8, Ts11, and Ts15)[6], and two double-trisomic (Ts6 + 8 and Ts8 + 15) EGFP/Luc-labeled ESC lines were established (Supplementary Fig. 2a, b). Whole-genome sequencing (WGS) and chromosomal counting of metaphase chromosomes confirmed the single and double trisomies, respectively (Supplementary Fig. 2c, d). We then performed subcutaneous injection of these cells into SCID mice, followed by surgical excision of the primary teratoma before it reached 1.5 cm in diameter, which allowed prolonged observation of potential metastasis. Tumor growth was monitored non-invasively by in vivo bioluminescence imaging (BLI) using an IVIS spectrum (Supplementary Fig. 3a).

Bioluminescence signals were detected ectopically in mice approximately 2–3 months after the injection (Fig. 1b). To define the precise location of metastatic lesions, ex vivo BLI was subsequently performed on the affected organs. Remarkably, bioluminescence signals were observed in multiple organs collected from the mice injected with trisomic ESCs, including the lung, liver, spleen, intestine and kidney (Fig. 1c and Supplementary Fig. 3b). Histologic examination of lung sections revealed obvious metastatic lesions in mice injected with trisomic ESCs (Fig. 1d and Supplementary Fig. 3c). Trisomic ESC-derived metastases were seeded near blood vessels and composed of many undifferentiated malignant cells (Fig. 1e), which were positively stained for GFP and pluripotency-associated factor OCT4 (encoded by *Pou5f1*) (Supplementary Fig. 3d, e). The origin of the metastases was further confirmed by genomic PCR amplification of the unique *PB* transposon-host junction fragments in each trisomic cell line (Fig. 1f). Metastasis generally (86.79%) occurred between 8- and 15-weeks post injection (Fig. 1g). The overall metastatic status of trisomic ESC-derived teratomas and successive analysis of metastases are summarized (Fig. 1h and Supplementary Tables 1, 2). In clear contrast, the mice injected with WT ESCs (34 mice) did not show any metastasis during 3 - 4 months of observation using the same monitoring procedures (Supplementary Figs. 1d,

3f, g). These findings are consistent with numerous studies conducted since the early 1980s in which teratomas formed by diploid ESCs have never been reported to metastasize[21].

### No additional CNVs or cancer driver gene mutations are required during teratoma metastasis

Metastatic spread is an evolutionary process that is often accompanied by the acquisition of gene mutations and chromosome copy number alterations[22]. To study whether additional copy number variations (CNVs) were acquired during trisomic teratoma metastasis, we performed WGS of primary teratomas and metastatic lesions derived from Ts6, Ts8, Ts11, Ts15, Ts6 + 8, and Ts8 + 15 ESCs, which revealed that most metastases presented CNV patterns similar to those of primary teratomas (Fig. 2a), while only two metastatic samples derived from Ts6 and Ts8 displayed additional single-chromosome gains (Supplementary Fig. 4 and Supplementary Fig. 5). To further evaluate whether cancer driver gene mutations accumulated during teratoma metastasis, we performed whole-exome sequencing (WES, 150×) of paired primary teratomas and metastases (CNVs of these metastatic samples were analyzed and showed in Supplementary Fig. 5). No common mutation was identified among different aneuploidies and no mutation in reported cancer driver genes was identified in any metastatic samples[23] (Fig. 2b–g). In reference to the primary teratoma, the number of gene mutations obtained during metastasis was quite low (a median of 20 variants) (Fig. 2e). These results indicate that no additional CNVs or cancer driver gene mutations are required during aneuploid ESC-derived teratoma metastasis.

### Metastatic phenotypes can be repressed by trisomy correction

To determine whether aneuploidy is the direct cause of teratoma metastasis, we investigated whether trisomy correction could rescue the metastatic phenotype. The strategy of trisomy correction was successfully applied by our group to obtain isogenic diploid ESCs in a previous study[6]. Briefly, FIAU (fialuridine) selectively killed cells harboring the *PB* transposon (carrying the HSV-*ΔTK* negative selectable marker) in the trisomic chromosome, while cells that have randomly lost the trisomic chromosome could survive. Using this strategy, Ts8 and Ts11 ESC lines were reverted to diploid cells (named Di8 and Di11) (Fig. 3a). Embryoid body (EB) differentiation experiment was used to evaluate in vitro differentiation potential of pluripotent stem cells[24,25]. We found that the compact EB morphologies of Ts8 and Ts11 ESCs were successfully rescued in Di8 and Di11 ESCs, shown as EBs with large cystic structures (Supplementary Fig. 6). Moreover, we performed teratoma experiments assessing metastasis using the strategy described above (Fig. 1a). After surgical excision of primary teratomas and prolonged observation for approximately 2- 3 months, neither Di8 (the number of mice with metastasis in total observed mice, Met/ Total = 0/6) nor Di11 (Met/Total = 0/7) teratoma-bearing mice showed metastasis (Fig. 3b, c). It was further supported by tail vein injection of Ts15-Met cells (isolated and cultured from metastatic sites) and its isogenic diploid metastatic cells (named Di15-Met), since the metastatic efficiency was significantly reduced in Di15-Met (Fig. 3d, e and Supplementary Fig. 7a–c). Thus, trisomy correction remarkably blocked teratoma metastasis. Next, we detected the CIN status in the injected cells by time-lapse imaging using a histone 2B-mCherry (H2B-mCherry) reporter. We assayed chromosome mis-segregation events from two aspects: time required for mitosis from envelope breakdown to anaphase onset, and the frequency of mis-segregation events including chromosome bridge, lagging chromosomes, multinucleated cells, and multipolar division. We found that the mitotic time and the mis-segregation events of trisomic cells were very close to WT (Supplementary Fig. 8 and Supplementary Movies 1–3). Taken together, these results demonstrate that aneuploidy is one of the contributing factors to teratoma metastasis.

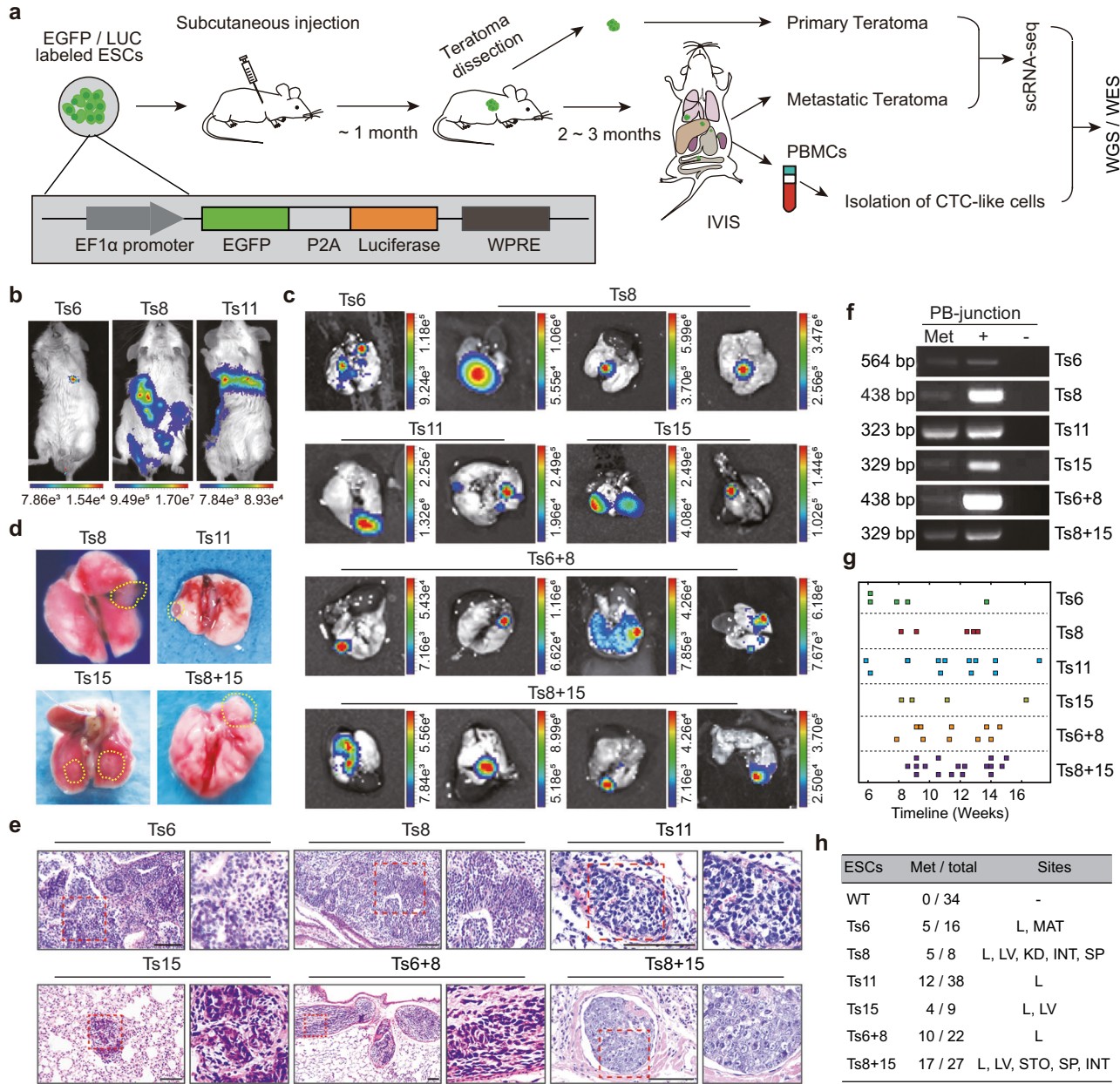

**Fig. 1 | Aneuploidy promotes the metastasis of teratomas derived from ESCs.**
**a** Schematic overview of the experiment. LUC, luciferase; PBMCs, peripheral blood
mononuclear cells; CTC-like cells, circulating tumor cell-like cells isolated from the
peripheral blood of aneuploid teratoma-bearing mice; scRNA-seq, single-cell RNA
sequencing; WGS, whole-genome sequencing; WES, whole-exome sequencing.
**b** Whole-body BLI imaging of aneuploid teratoma-bearing mice. **c** Ex vivo BLI
imaging of lung metastases from mice bearing aneuploid teratomas. The rainbow
gradient bar represents the photon flux. **d** Images of lung metastases. The yellow
dotted lines indicate metastatic lung lesions. **e** HE analysis of metastatic foci. Scale
bars, 100 μm. **f** Confirmation that metastases were derived from the injected ESCs
by genomic PCR. Histologic analysis of (**e**) and (**f**) are reproducibility available.
Source data are provided as a Source data file. **g** Timeline of metastatic events
observed in mice bearing different aneuploidies. Ts, trisomy. **h** Summary of
aneuploid ESCs-derived teratoma metastasis. Met/total, the number of metastatic
mice in the total observed mice. Sites, metastatic sites. L, lung; MAT, mediastinal;
LV, liver; KD, kidney; INT, intestine; SP, spleen; STO, stomach. See also Supple-
mentary Tables 1 and 2.

## CTC-like cells have high capacities for migration and organ colonization

Circulating tumor cells (CTCs) can survive in the bloodstream and seed
metastases[26,27]. After peripheral blood mononuclear cells (PBMCs)
were collected from aneuploid teratoma-bearing mice, we successfully
isolated CTC-like cells from mice bearing Ts6 + 8 teratomas and
established two cell lines (Fig. 4a). The origin of CTC-like cells was
confirmed by PCR and eGFP expression (Supplementary Fig. 9a, b).
Immunofluorescence (IF) staining revealed that these CTC-like cells
showed high Oct4 expression, suggesting that they possessed

"stemness" traits, although their morphology differed significantly
from that of ESCs (Fig. 4a and Supplementary Fig. 9c). Analysis of CNVs
in the cultured CTC-like cells confirmed the conserved CNV patterns
with Ts6 + 8 ESCs (Fig. 4b). Transwell assays revealed that CTC-like
cells exhibited increased invasive and migratory behavior in vitro in
comparison with WT ESCs (Fig. 4c). To assess the in vivo colonization
capacity of CTC-like cells, we transplanted $5 \times 10^5$ cells into immuno-
deficient mice via tail vein injection and found metastatic lesions in
multiple organs within one month (Fig. 4d, e, and Supplementary
Fig. 9d). These results suggest that metastatic dissemination in

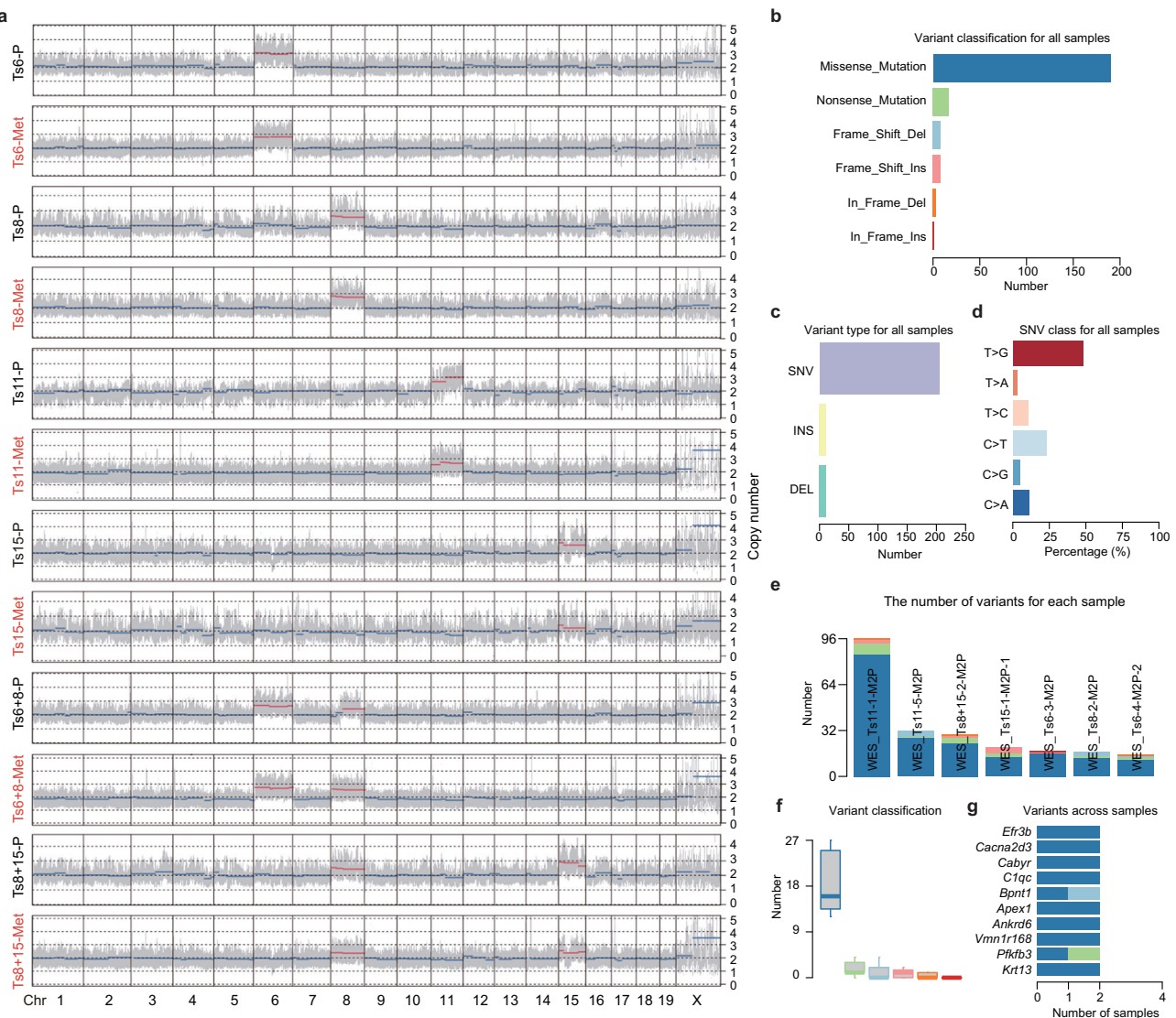

**Fig. 2 | Metastasis requires no additional CNVs or cancer driver gene mutations.** **a** Dot plot showing the CNVs imputed by WGS in primary teratoma (P, *n* = 6 biological independent samples) and metastatic (Met, *n* = 6 biological independent samples) samples. WT ESCs were used as the control. The gray bars indicate the interquartile ranges. The thick horizontal lines indicate segmented values. Trisomic chromosomes are indicated in red. Chr, chromosome. WES revealed the somatic mutations arising during metastasis (**b**–**g**, *n* = 7 biological independent samples).

Bar chart displaying the total number of variant classifications (**b**), variant types (**c**), and SNV classes (**d**) identified in metastatic samples. **e** Bar plot showing the variant number and variant classifications detected in each metastatic sample. **f** Box plot showing the number of variant classifications. Data are median (line within the box) with a SD (box limit) and 1.5 × outliers (whiskers). **g** Bar plot showing the frequency of gene mutations in metastatic samples. Top 10 mutated genes were presented. Colors were used to represent the variant classification (**b**, **e**–**g**).

trisomic teratoma-bearing mice occurs through the blood circulation and that CTC-like cells show a high capacity for migration and organ colonization.

## Stem cell traits are enriched in metastatic lesions

To explore the molecular mechanisms underlying aneuploid teratoma metastasis, we performed single-cell RNA sequencing (scRNA-seq) analyses of six lung metastases derived from Ts6, Ts8, Ts11, Ts6 + 8, and Ts8 + 15 ESCs (designated Ts6-M, Ts8-M, Ts11-M, Ts6 + 8-M, Ts8 + 15-M1 and Ts8 + 15-M2) and seven primary teratoma samples (designated WT-P, Ts6-P, Ts8-P, Ts11-P, Ts15-P, Ts6 + 8-P and Ts8 + 15-P), respectively. We also performed scRNA-seq of WT and four single-trisomic ESC lines (designated WT-C, Ts6-C, Ts8-C, Ts11-C, and Ts15-C). A total of 147,275 high-quality cells were acquired and partitioned into 16 clusters with distinct transcriptome characteristics (Fig. 5a and Supplementary Fig. 10a, b). Based on the expression of *EGFP*, *luciferase* and pluripotent markers such as *Pou5f1* and *Dppa5a*, eight clusters of ESC-derived cells

were extracted (Fig. 5b). The CNV prediction results further confirmed the identification of ESC-derived clusters (Supplementary Fig. 10c). The other eight clusters (granulocyte, macrophage, DC, ILC, erythroid, endothelium, fibroblast, and epithelium; see Supplementary Fig. 10b and Supplementary Table 3 for the annotation markers) came from SCID mice and were excluded from subsequent analyses.

Depending on the expression of *Pou5f1* and *Dppa5a*, the ESC-derived clusters could be further divided into two stemness states: ES_Ori and ES_Stem, and six differentiated states: ES_Musc (highly expressing *Des*, *Tnnc2*, *Myl1*), ES_NSC (highly expressing *Nes*, *Msi1*), ES_Oligo (highly expressing *Olig1*), ES_Schw (highly expressing *Sox9*, *Gfap*), ES_NC (highly expressing *Tubb3*, *Map2*), and ES_FB (highly expressing *Col1a1*, *Col3a1*) (Supplementary Fig. 10b and Supplementary Table 3). As expected, ES_Ori cells, which showed the highest expression levels of pluripotency genes, were the major component of the cell line samples, whereas in primary teratomas and metastases, differentiated ESC-derived clusters were observed. Notably, in

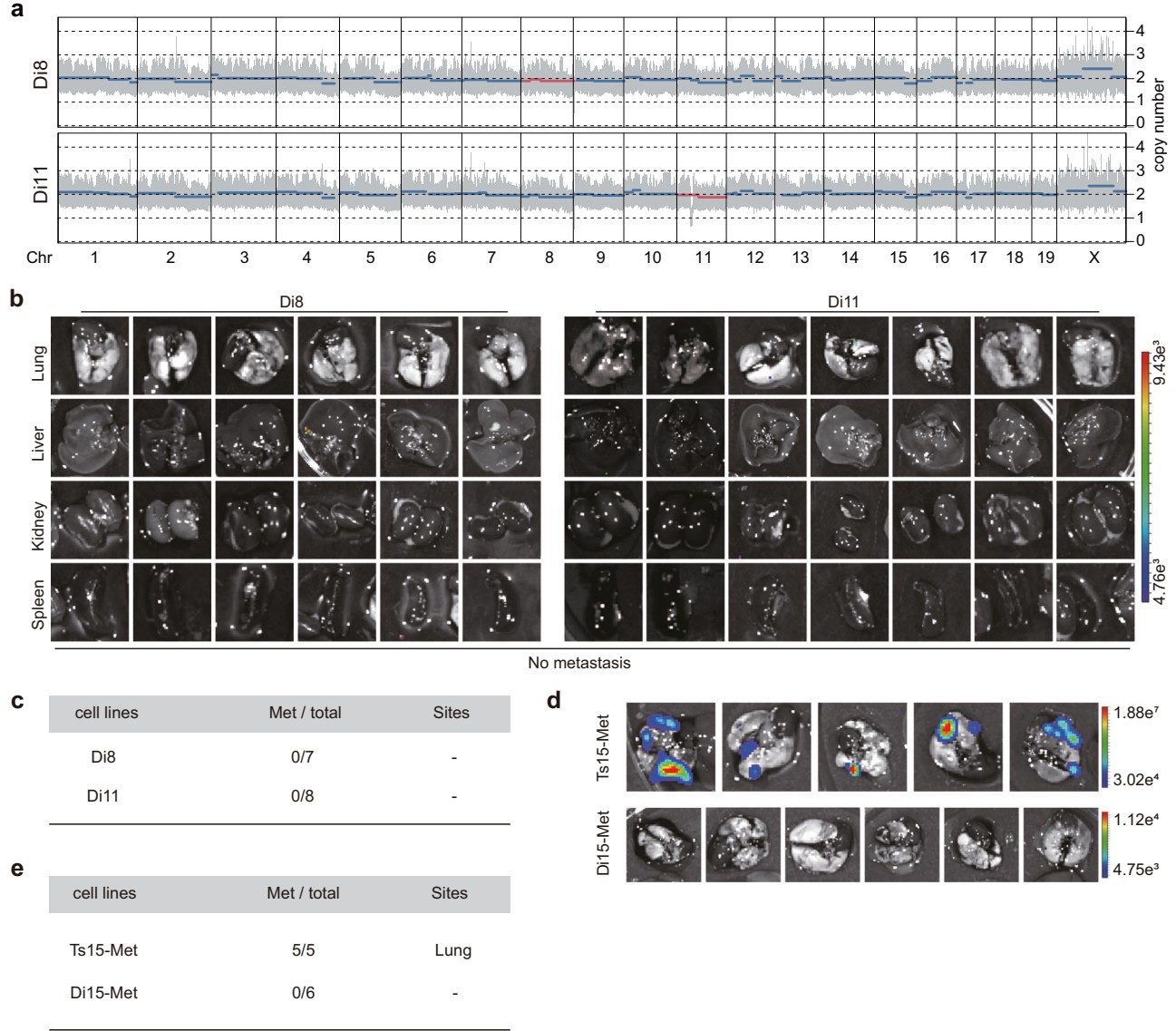

**Fig. 3 | Trisomy correction rescued the metastatic phenotypes of aneuploid ESCs. a** Chromosome plot indicating CNVs of isogenic diploid cell lines derived from Ts8 and Ts11 (designated Di8 and Di11, respectively). Di, Diploidy. WT ESC was used as a control. CNVs were analyzed using Sequenza. A gray bar indicates the interquartile range. The thick horizontal lines indicate segmented values. Rescued trisomic chromosomes are indicated in red. Chr, chromosome. **b** BLI imaging of Di8 and Di11 teratoma-bearing mice after 2–3 months of injection ($n = 6$). Left,

ex vivo BLI images of organs from Di8. Right, ex vivo BLI images of organs from Di11 ($n = 7$). Di, diploidy. The rainbow gradient bar shows the range of photon flux. **c** Summary of metastatic efficiency of Di8 and Di11. **d** BLI imaging of SCID mice injected with Ts15-Met ($n = 5$) and Di15-Met ($n = 6$) cells lasted for about 1 month. Di, diploidy. The rainbow gradient bar shows the range of photon flux. **e** Summary of metastatic efficiency of Ts15-Met and Di15-Met.

contrast to WT teratomas, which were fully differentiated one month after injection, a substantial number of cells with intermediate levels of pluripotency marker expression (ES_Stem) still existed in aneuploid teratomas. Of particular interest, compared with aneuploid teratomas, the proportions of ES_Stem cells were increased further in aneuploid metastases, and in some cases, we even observed ES_Ori cells (Fig. 5c). We speculated that stemness maintenance might play a critical role in promoting teratoma metastasis.

Cell trajectory analysis was performed on eight clusters to reconstruct the differentiation trajectory from ESCs to primary teratomas and to metastases. ES_Stem was identified as the hub connecting ES_Ori and differentiated clusters (Fig. 5d). Importantly, we noted that many proteasome subunit genes were continuously downregulated along with the differentiation route (Fig. 5e). We then compared the differentially expressed genes between ES_Stem and ES_Ori and confirmed the above results (Fig. 5f and Supplementary Fig. 10d).

This suggests that the downregulation of proteasome activity underlies the stemness maintenance in aneuploid ESCs-derived teratomas and metastases.

## Aneuploid embryoid bodies exhibited proteasome dysfunction and overactivated ER stress

Proteasome activity was expected to be elevated in the early stage of ESC differentiation to degrade damaged proteins, including carbonylated and advanced glycation end proteins[28,29]. However, no transient upregulation of proteasome subunit genes was observed along the pseudotime trajectory during aneuploid ESCs differentiation (Fig. 5e). By using the embryoid body (EB) formation assay, we have previously found the differentiation timing of aneuploid ESCs to multiple lineages was delayed[6]. We then quantified the proteasome activity in different stages of WT and aneuploid ESCs differentiation in vitro. Unlike WT cells, the total proteasome activity in aneuploid cells quickly

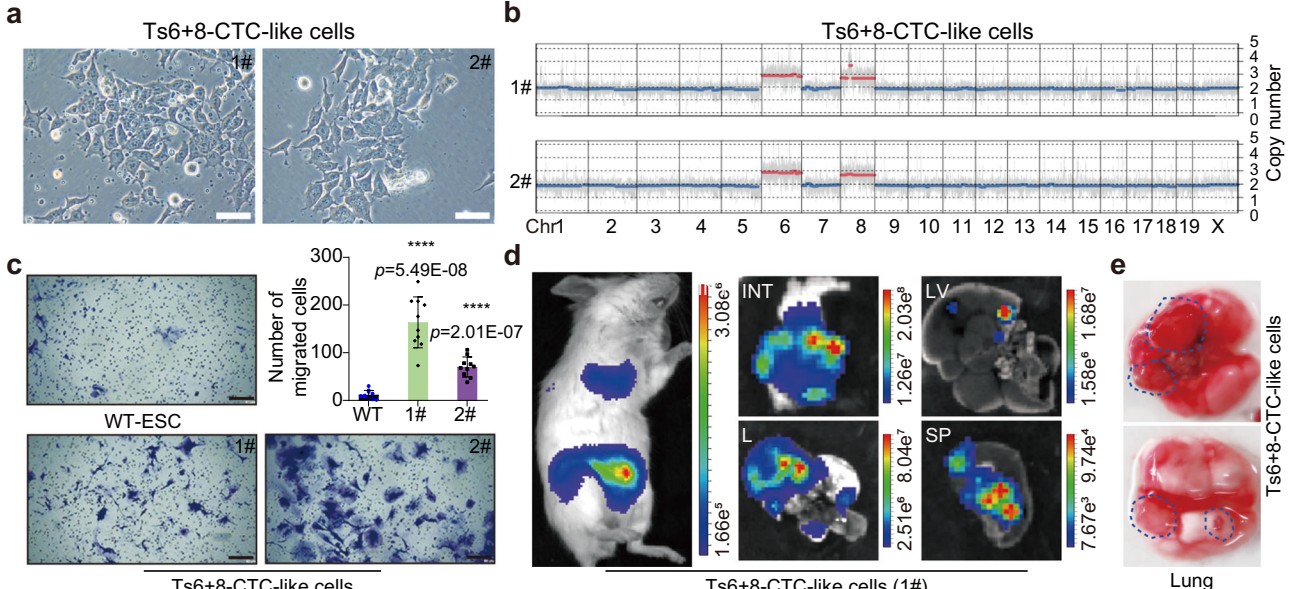

**Fig. 4 | Characteristics of cultured CTC-like cells. a** Two CTC-like cell lines (1# and 2#) were successfully established from the peripheral blood of Ts6 + 8 teratoma-bearing mice. Scale bars, 50 μm. Micrograph photos of CTC-like cell lines are reproducibility available. **b** CNV patterns of cultured CTC-like cells (1# and 2#). **c** Ts + 8 CTC-like cells (1# and 2#) exhibited an enhanced invasion ability compared with that of WT ESCs according to Transwell assays. Scale bars, 200 μm; For each group, $n = 10$. Error bars, ± SD. $P$ values were calculated using a two-tailed $t$ test. Source data are provided as a Source data file. **d** BLI imaging signals showed the CTC-like cells (1#) colonized multiple sites ($n = 5$). The rainbow gradient bar represents the photon flux. **e** Metastatic lung lesions were observed one month after the tail vein injection of CTC-like cells (1#). Blue dotted lines indicate metastatic nodes.

declined at day 2 of EB formation (Supplementary Fig. 11a). Accordingly, many carbonyl proteins accumulated in aneuploid cells and could not be degraded efficiently in the early stage of differentiation (Supplementary Fig. 11b, c). The protein levels of proteasome subunits PA28α and β5i were also found to be lower than those in WT cells (Supplementary Fig. 11d, e). The accumulation of misfolded and damaged proteins could trigger endoplasmic reticulum (ER) stress, and the unfolded protein response (UPR) is activated to regain ER protein homeostasis[30,31]. We therefore analyzed activation of the UPR pathway by detecting three canonical branches of UPR (IRE1, PERK, and ATF6) during EB differentiation. We found the phosphorylation of PERK (p-PERK) and phosphorylation of EIF2A (p-EIF2A) were elevated in trisomic cells. Similarly, processed active ATF6 (50 kDa) and its transcription target chaperone protein BIP was significantly increased in most trisomic cells on differentiation day 2 or day 4 (Supplementary Fig. 11f, g).

We further performed bulk RNA-seq of differentiated EBs (Day 8), and found that aneuploid EBs harbor a higher ER stress score (GO: 0036493) compared with WT EBs (Supplementary Fig. 11h). Besides, aneuploid EBs displayed higher level of stemness (Ts6, Ts8, Ts15) (Supplementary Fig. 11i) which is generally consistent with scRNA-seq data, and existed differentiation defects in mesoderm (Ts6, Ts8, Ts15), and endoderm (Ts6, Ts8, Ts11, Ts15) (Supplementary Fig. 11j–l). These results suggest that insufficiency of proteasome activity and overactivated UPR might underlie the differentiation defects of aneuploid cells.

We then treated aneuploid cells with proteasome activator Oleuropein[32] during EB differentiation, and found that Oleuropein treatment could increase the proteasome activity in aneuploid cells to some extent, and promoted aneuploid EB formation with large cystic structures (Supplementary Fig. 11m, n).

### The proteasome activator and endoplasmic reticulum stress pathway inhibition could inhibit the metastasis of aneuploid teratomas

We then administrated Oleuropein to SCID mice once the teratomas were palpable (Fig. 6a), and found that treatment with Oleuropein effectively reduced aneuploid teratoma volume (Supplementary Fig. 12a and Source data file) and partially rescued the deficiency of aneuploid ESC differentiation, as evidenced by significantly decreased proportions of OCT4+ cells in the teratoma (Fig. 6b). Approximately 2- 3 months after Oleuropein administration, none of Ts8, Ts11 or Ts8 + 15 teratoma-bearing mice showed metastasis while the vehicle (saline) showed strong metastatic signals (Figs. 6c, 6d and Supplementary Fig. 12b, c). Thus, Oleuropein can remarkably repress the metastatic spread of aneuploid cells. Similarly, ER stress inhibitor 4-PBA[33] effectively inhibited the metastasis of aneuploid teratomas as well (Supplementary Fig. 13). We further explored the effects of UPR inhibition by generating UPR genes (*Atf6*, *Xbp1*, *Eif2α* target three branches of UPR, respectively) knockdown metastatic cells of Ts15-Met and Ts8 + 15-Met via CRISPR interference[34] (CRISPRi) (Supplementary Fig. 14). The metastasis was routinely monitored by BLI after tail vein injection of Ts15/Ts8 + 15-Met cells (Fig. 6e). We found Ts15-Met and Ts8 + 15-Met cells with UPR gene knockdown exhibited decreased metastasis efficiency compared with control (Fig. 6f–i). Taken together, these results suggest that both activation of proteasome activity and repression of ER stress could prevent the metastasis of aneuploid teratomas.

## Discussion

Compared to other widely used aneuploid models[7–10,35–37], the ESC-derived teratoma model is unique in that it allows the role of aneuploidy in metastasis to be investigated. Normal-karyotype mammalian ESCs readily form mature teratomas with three germ layers after transplantation, and aneuploid ESCs and iPSCs tend to form immature teratomas with rapid proliferation and reduced differentiation; neither of these teratoma types has ever been observed to metastasize[6,13,21,38]. In this work, we have shown that aneuploidy alone is able to drive the metastasis of ESC-derived teratomas via the retention of stemness and have discovered a cluster of stem cells (ES_Stem) that might seed teratoma metastasis. We have further demonstrated that insufficiency of proteasome activity and over-activated UPR might underlie the differentiation defects of aneuploid

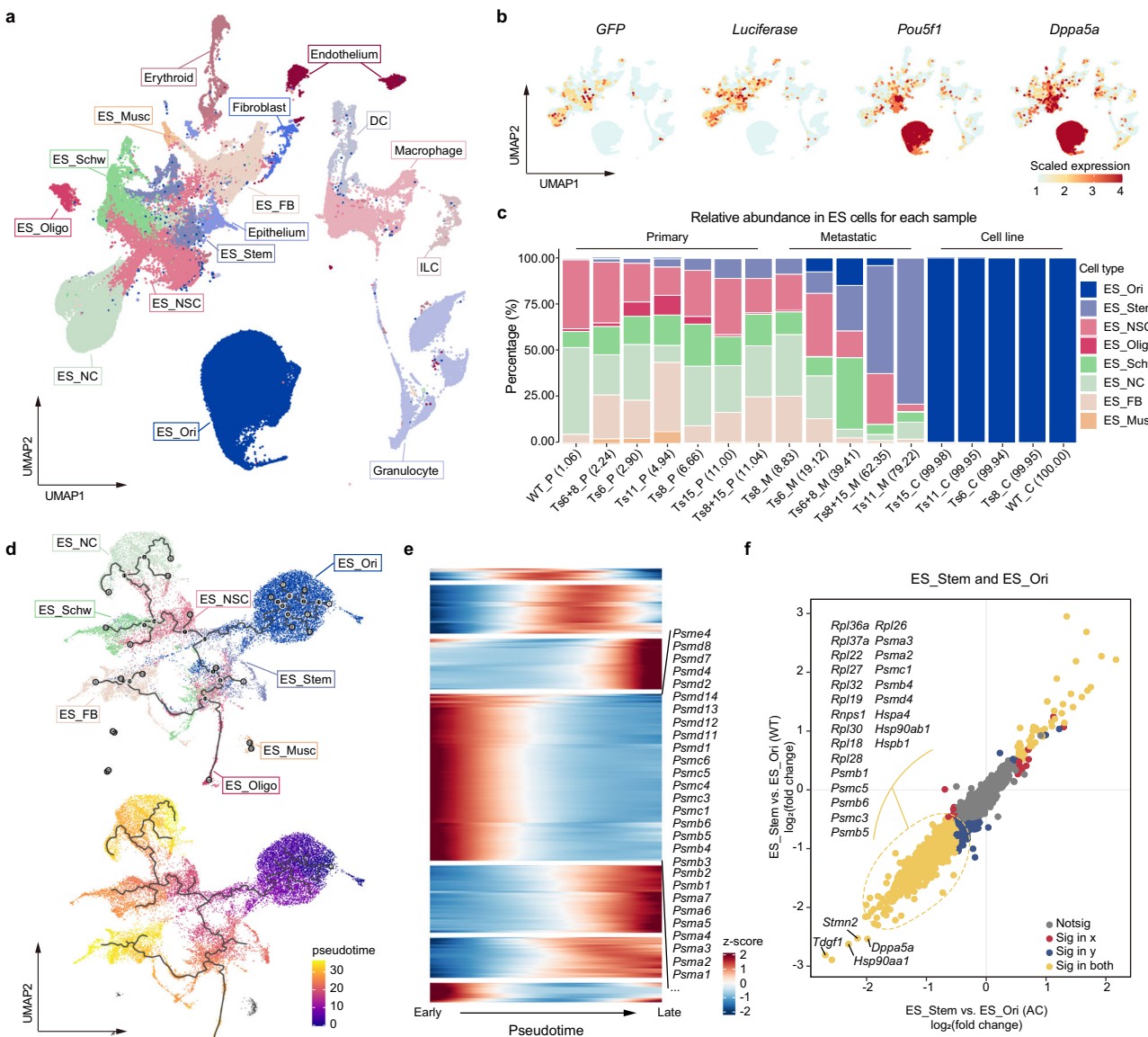

**Fig. 5 | scRNA-seq analysis identifies stem-like aneuploid cell clusters in metastatic samples. a** UMAP projection of scRNA-seq data. DC, dendritic cells. ILCs, innate lymphoid cells. ES, derived from aneuploid or WT cells. ES_Ori, original ESCs. ES_Stem, stem cell population. ES_NSC, neural stem cells. ES_Oligo, oligodendrocyte. ES_Schw, Schwann cell. ES_NC, neuron cells. ES_FB, fibroblast cells. ES_Musc, Muscle cells. **b** Expression of *EGFP* and *luciferase* in *EGFP-luciferase* dual-labeled cells and expression of *Pou5f1* and *Dppa5a* in ESCs. **c** The proportion of stemness (ES_Ori and ES_Stem) is depicted in different samples, including cell lines, primary teratomas, and metastases. **d** Pseudotime analysis-reconstructed differentiation routes (branches and nodes) of different cell types recognized by scRNA-seq data analysis. **e** Heatmap plot displaying the expression of specific genes (listed on the right) associated with the proteasome that were consistently decreased among various differentiation stages (from early to late) across the WT and aneuploid cells in pseudotime order (**d**) (scaled by row). **f** Differentially expressed genes in ES_Stem compared with ES_Ori (split into WT and AC). AC, aneuploid cells.

---

stem cells, and targeting these pathways can inhibit aneuploid teratoma metastasis.

Previous study reported a complicated relationship between aneuploidy and metastasis by using single-chromosome aneuploidies derived from the human colon cancer cell line HCT116. They found 12 out of 13 specific aneuploidies suppressed or were neutral to metastasis-related processes, but chromosome 5 trisomy exhibited the metastasis-promoting effect[11]. Yet, aneuploidy is frequently associated with poor prognosis of cancer patients[20,39,40]. Another study demonstrated that CIN, rather than aneuploidy, drives metastasis by triggering the cGAS/STING cytosolic DNA response[5]. In our study, the trisomic ESCs, primary teratomas, and metastases showed no obvious chromosome number alterations, while trisomic ESC-derived teratomas always disseminated to multiple distant organs. Our findings

are consistent with the clinical observations and render the relationship between aneuploidy and metastasis more clear. Aneuploidies are universal teratoma metastasis promoters, irrespective of the identity of the specific extra chromosomes. Our study favors chromosome copy-number changes shape tumor evolution and metastasis[16].

The proteasome system plays an important role during ESC differentiation[28,29]. We found that proteasome activity could not be sufficiently stimulated to ensure the differentiation of aneuploid ESCs. Insufficient proteasome activity results in protein redundancy in aneuploid cells, which induces excessive UPR. UPR plays an important role in controlling cell fate decisions and is associated with differentiation[31], while excessive UPR blocks in vitro blastocyst development[41]. Here, we used the proteasome activator oleuropein or the ER stress inhibitor 4-PBA to aid proper differentiation of aneuploid

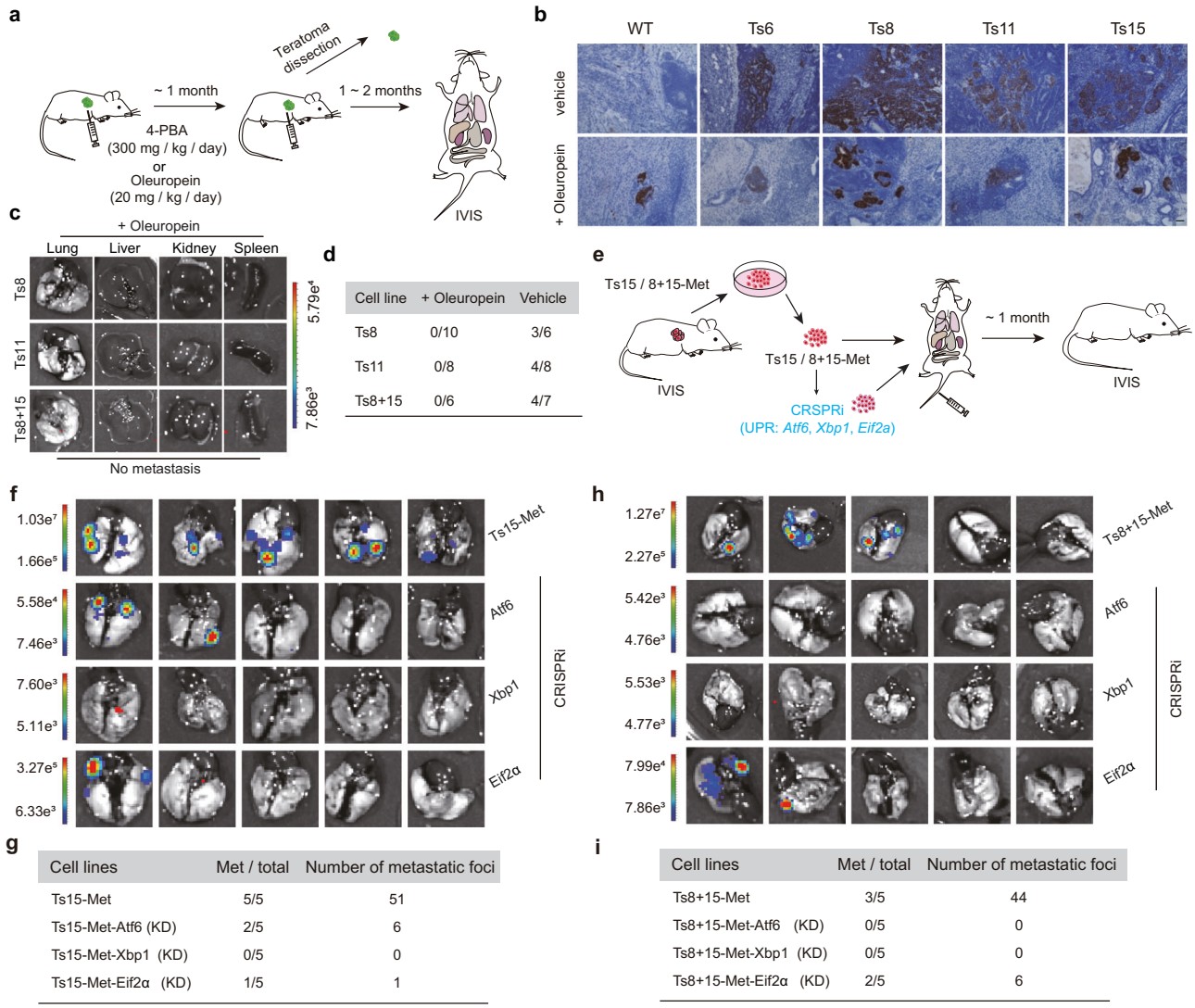

**Fig. 6 | Aneuploidy-induced metastasis can be inhibited by proteasome activator treatment and UPR inhibition. a** Schematic overview of the drug treatment experiment. **b** IHC experiment detecting OCT4 expression in vehicle- or Oleuropein-treated WT and aneuploid primary teratomas (n = 3). Vehicle, control group treated with saline. Scale bar, 50 μm. Histologic analysis of vehicle- or Oleuropein-treated WT and aneuploid primary teratomas are reproducibility available. **c** BLI imaging of distant organs from teratoma-bearing mice of different aneuploidy exposure to proteasome activator Oleuropein. The rainbow gradient bar represents the photon flux. **d** Summary of metastatic efficiency in teratoma-bearing mice treated with Oleuropein or vehicle among different aneuploidies. **e** Schematic overview of the tail vein injection experiment. **f** BLI imaging of distant organs from mice injected with primary metastatic cell lines Ts15-Met or UPR genes knockdown derivants (via CRISPRi). The rainbow gradient bar represents the photon flux. **g** Summary of metastasis from Ts15-Met. **h** BLI imaging of distant organs from mice injected with primary metastatic cell lines Ts8 + 15-Met or UPR genes knockdown derivants (via CRISPRi). The rainbow gradient bar represents the photon flux. **i** Summary of metastasis from Ts8 + 15-Met.

ES cells. Oleuropein or 4-PBA suppressed metastasis of aneuploid ESC-formed teratomas in SCID mice. The activation of proteasome activity by oleuropein might be an important mechanism of suppressing teratoma metastasis, while the volume reduction of the aneuploid teratomas after treated with oleuropein might be one of the contributing factors as well. Moreover, silencing critical components of UPR by CRISPRi significantly inhibited metastasis of the secondary transplant of Ts15 and Ts8 + 15. In fact, interfering with the UPR is a potential strategy for cancer treatment[30,33,42]. Our study provides evidence of alleviating proteotoxic stress or ER stress in aneuploid cells as the targets of cancer therapy.

CTCs are extremely rare in peripheral blood (-1-100 cells per ml)[43–45]. Of them, only a small part can be captured from the peripheral blood mononuclear cells. The isolation of viable CTCs from murine blood is even harder. Most CTCs isolation methods utilize the unique surface marker (EpCAM) of CTCs[46,47], as well as the physical properties of CTCs including their size and density[48]. The low frequency of CTCs, the diminished cell viability and the heterogeneity observed in CTCs make the isolation of a pure population of CTCs laborious. Successful culture of the specific type of CTCs relies on the culture conditions. Here, we are lucky to successfully isolate and culture CTC-like cells with stem cell traits from mice bearing Ts6 + 8 teratomas under the culture conditions of ESCs. However, CTCs in other states will probably not be isolated under the conditions we used. So, CTC-like cells isolated and cultured in vitro are less likely to represent the heterogeneity of CTCs.

Most cancers harbor both aneuploidy and oncogenic mutations[3]. However, aneuploidy has been shown to be detrimental to cell fitness in the context of somatic cells[10,49,50]. Meanwhile, we and others have recently reported that in morphologically normal human tissues substantial somatic mutations in cancer driver genes can accumulate, but copy number variations/aneuploidy are rarely seen[51–53]. Our findings

presented here highlight the oncogenic role of aneuploidy and imply a synergistic effect between gene mutations and aneuploidy in tumor progression and metastasis.

## Methods

### ES cell culture

Aneuploid ESC lines were derived from murine AB1 129 ESCs (WT). WT and aneuploid ESCs were cultured on γ-irradiated MEFs (iMEFs) in M15L medium composed of knockout DMEM (Gibco), 15% fetal bovine serum (HyClone), 1% GlutaMAX (Gibco), 0.67% nonessential amino acids (NEAA) (Gibco), 0.5% penicillin–streptomycin (Gibco), 100 μM β-mercaptoethanol (Gibco), and 1000 U/mL recombinant mouse leukemia inhibitory factor (LIF) (Millipore). Cells were cultured at 37 °C in a humidified environment containing 5% $CO_2$. Cells were passaged routinely every other day and dissociated with 0.05% trypsin-EDTA (Gibco, 25300-062) at 37 °C for 4 min. The medium was changed every day.

### Lentivirus preparation and cell labeling with EGFP-Luciferase

Plasmids carrying EGFP-P2A-Luciferase elements were purchased from Obio Technology (Shanghai). Lentiviruses were generated using the 2nd generation lentiviral packaging plasmids (PSPAX.2 and PMD2.G, the gift of Zhang lab) and packaged in Lenti-X 293T cells (Clone Tech, a gift of Huang lab). Lentivirus condensation was performed using the Ultrafiltration tube (100 KDa, Millipore). First, the tubes were balanced with 15 mL DPBS and centrifuged at 5000 × $g$ for 15 min, after that lentivirus (15 mL) was centrifuged at the same speed for another 30 min at 4 °C. Finally, the condensed lentivirus was used for lentivirus transduction when cells were passaged. The system was consisted of 1.4 mL M15L medium, 1.5 μL polybrene (8 mg/mL), and 100 μL condensed lentivirus. Lentivirus transduction was repeated on the next day. After 72 h, cells were collected for FACS to sort GFP-positive cells.

### Teratoma experiments

Male SCID/Beige mice (6 weeks old) were purchased from Charles River Laboratories (Wilmington, MA). All animal experiments were approved by the Institutional Animal Care and Use Committee of the Chinese Academy of Medical Sciences & Peking Union Medical College. All animal care and experimental methods followed the ARRIVE guidelines for animal experiments. Mice were randomly grouped by body weight. After sedimentation for 30 min at 37 °C to remove iMEFs, 1–2 × $10^5$ ESCs in 100 μL DPBS suspension were subcutaneously injected into one side of the dorsal flanks of SCID/Beige mice. Approximately 3–4 weeks after injection, when the tumor diameter reached 1.5 cm, primary teratomas were removed. Mice that survived allowed the prolonged observation of metastasis. For whole-genome sequencing, primary teratomas were collected and stored at −80 °C after fresh freezing in liquid nitrogen. For hematoxylin-eosin (HE) staining and immunohistochemical (IHC) staining, primary teratomas were partially sectioned and fixed in 10% buffered formalin phosphate for 24 h, after which they were embedded in paraffin for tissue sectioning (Bioservice).

### EB differentiation

When WT and aneuploid ESCs reached 85–90% confluence, they were dissociated with trypsin (0.05%) and sedimented for 30 min at 37 °C, after which 3 × $10^7$ ESCs were transferred to low-attachment 150 mm-diameter bacteriological-grade Petri dishes in EB formation medium (M15 medium). The culture medium was replaced with fresh EB formation medium every 2 days for 9–10 days.

### Bioluminescence imaging

Approximately one to two months after primary tumor excision, mice were observed by IVIS (PerkinElmer). Mice were placed under general anesthesia via the intraperitoneal injection of sterile tribromoethanol (Sigma) or isoflurane inhalation (YIPIN PHARMACEUTICAL). Next,

200–300 μL of luciferin (PerkinElmer, 15 mg/mL) was administered to the mice via intraperitoneal injection. Then, relapsed or residual primary teratomas were removed to reduce background signal interruption, and the mice were imaged to allow the analysis of metastasis at distant organs. The mice were sacrificed and dissected, and ex vivo organ bioluminescence imaging was performed immediately to confirm the metastatic foci. Some distant organs with metastatic foci were flash-frozen with liquid nitrogen or stored in frozen stock solution (provided by CapitalBio Technology) on ice for further sequencing, and others were fixed in 10% buffered formalin phosphate for 24 h, followed by histological analysis.

### Histological analysis

For HE staining, slides were deparaffinized twice in xylene (5 min each), hydrated, stained with hematoxylin/eosin, and dehydrated using a series of ethanol concentrations. The slides were mounted and scanned to obtain whole-section images, which were analyzed using NDP.viewer Software from Hamamatsu, Caseviewer, or ImageScope. Pathological analysis was mainly carried out by the Department of Pathology at Peking Union Medical College Hospital. For IHC staining, briefly, samples were sectioned (5 μm), dewaxed in xylene, dehydrated using a series of alcohol concentrations, and covered with 3% $H_2O_2$ for 30 min at room temperature to block endogenous peroxidase activity. Next, the samples were rinsed three times with PBS (5 min each) and treated with citrate buffer (pH 6.0) at 121 °C for 5–7 min for antigen retrieval. After blocking with goat serum for 30 min at room temperature, the slides were incubated with anti-Oct4 (1:100) and anti-GFP (1:100) overnight at 4 °C. Secondary antibodies were added to the slides, followed by processing with DAB chromogen. After staining and differentiation with hematoxylin, the slides were dehydrated in a series of alcohol concentrations and placed in xylene to allow the tissue sections to become transparent. The slides were mounted and visualized using a standard light microscope (Zeiss) at 100× magnification, or the whole section was scanned at 20–40× magnification (Nanozoomer, Hamamatsu, or Leica).

### Genomic PCR and nested PCR

Primer design and PCR were performed as described in our previous work[54]. Briefly, during the isolation of trisomic ESCs, the *piggyBac* (*PB*) transposon was inserted into the specific sites of ESC genome. The *PB* transposon-host junction fragments could be amplified by genomic PCR using PB5′-2 or PB3′-2 primers and the primers near the insertion sites. For micro-metastases from which genomic PCR yielded products with low concentrations, we designed nested PCR primers (Supplementary Table 4), and used the products from the first-round PCR amplification as templates for the second-round PCR.

### Chromosome counting of metaphase cells

Mouse ESCs and CTC-like cells were cultured in M15 medium for approximately 2 days until they were 70–80% confluent, after which they were treated with colcemid (Sigma, D1925) at a concentration of 0.2 μg/mL for 2–3 h at 37 °C. Metaphase spreads were prepared. Images of 40–200 metaphase spreads were captured by microscopy using a Zeiss Imager M2 with SmartCapture 3 software.

### Time-lapse imaging

Cells transfected with histone 2B-mCherry (H2B-mCherry) reporter were cultured on glass base dish (Thermo) for 24–48 h to achieve a 50–60% confluent. Glass base dish surface was pre-treated by Matrigel (BD). After cell surface was found, time-lapse imaging was performed by using Cell Discover 7 (CD7, Zeiss). Images were captured each 2 min (WT, Ts6, Ts8, Ts11) or 3 min (Ts8 + 15) lasted for 20–24 h and 9 scenes were set. Then we analyzed the mis-segregation events like chromosome bridge, lagging chromosomes, multinucleated cells and multipolar division of each cell line to count mis-segregation rate between aneuploid mouse ESC lines and WT (euploid mouse ESC line).

## Capture and isolation of CTC-like cells

To avoid the contamination from teratoma surgery, we isolate CTC-like cells from teratoma-bearing mice that did not undergo teratoma surgery. After peripheral blood was collected from SCID/Beige mice, the samples were slowly transferred into Ficoll to allow isolation of peripheral blood mononuclear cells (PBMCs). Blood samples were centrifuged at 2000 rpm for 10 min, and PBMCs were collected using 200 μL micropipettes or glass capillary tubes. Erythrocytes were removed after treatment with ACK lysis buffer (Biroyee) for 2 min in triplicate. Cells were washed with DPBS and then plated for cell culture using M15L medium. Approximately twenty days later, clones propagated in 6-well plates were passaged and identified. The M15L medium was changed every 4–5 days to prevent disruption of the clonal proliferation of CTC-like cells mixed with PBMCs. CTC-like cells images of bright field and eGFP were then captured using a fluorescence microscope (Zeiss) at 100× magnification.

## Transwell assay of CTC-like cells

Transwell chambers (8 μm; Corning, 3422) coated with 50 μL Matrigel (BD, 356234) (1 mg/mL) were treated in a 37 °C incubator for 4 h. After M2L medium was added to the upper and lower chambers and allowed to equilibrate for 1 h, the M2L medium was discarded, and $6 \times 10^4$ cells (WT, aneuploid mouse ESCs (Ts6 + 8) or CTC-like cells cultured from peripheral blood) that were FBS starved for 12 h were placed in the upper chamber. M2L medium was added to the upper chamber, and M15L medium was added to the lower chamber. After 36 h, the chamber was fixed with a 4% paraformaldehyde solution (PFA, Sigma) in a new 24-well plate for 10 min at room temperature. Next, the chamber was placed in another new 24-well plate with 500 μL ethanol and fixed for 10 min at room temperature. The chamber was washed twice with 50 mL DPBS and stained with a 0.1% crystal violet stain solution (Sorlabio) for 30 min at room temperature. The upper-chamber cell layer was successively rubbed away with a wet cotton swab and dry cotton swabs. Ten images of each cell line were captured using a standard microscope (Leica DMi8) at 50× magnification, and the cells were counted to determine the number of migrated cells.

## Immunofluorescence of CTC-like cells

CTC-like cells were seeded on 12-mm coverslips pre-treated with laminin (0.1 mg/mL) and cultured in M15L medium for 24 h. The cells were then fixed with 4% PFA for 15 min and treated with 0.1% Triton X-100 to increase cell membrane permeability. The cells were subsequently blocked with goat serum for 30 min at room temperature, stained with anti-Oct4 (1:100), and incubated in a humidifying box at 4 °C overnight. Next, the secondary antibody was added to the slides, and they were incubated in a humidifying box at room temperature for 1 h, followed by DAPI staining for 30 min. Images were captured using a fluorescence microscope (Zeiss) at 100× magnification.

## Isolation and culture of metastatic cells

To focus on the precise metastatic sites, mice were sacrificed after IVIS imaging and ex vivo organ bioluminescence imaging was performed immediately to confirm the metastatic foci. The samples were digested using the Tumor Dissociation Kit (Miltenyi Biotec, Catalog no. 30-096-730) with a Gentle MACS Octo Dissociator (Miltenyi Biotec, USA) according to the manufacturer's protocol. Erythrocytes were removed after treatment with ACK lysis buffer (Biroyee) for 2 min in triplicate. Cells were washed with DPBS and then plated for cell culture using M15L medium. Clones propagated in 6-well plates were passaged and cryopreserved.

## UPR genes knockdown

CRISPRi system used in our screens relies on the catalytically inactive Cas9 (dCas9) protein fused with a KRAB repressor domain (dCas9-KRAB), targeted through single-guide RNAs (sgRNAs), to the specific loci upstream of promoters' region. CRISPRi sgRNAs were designed via CRISPick (Broad Institute) and selected by NCBI-primer blast. Briefly, sgRNA sequence for Ts15/Ts8 + 15-Met: *Atf6*-F1: CACCGACACCTCT CCCTCACAACT; F2: AAACAGTTGTGAGGGAGAGGTGTC. *Xbp1*-F1: CACCTAGACGTTTCCTGGCTATGG; F2: AAACCCATAGCCAGGA AACGTCTA. *Eif2a*-F1: CACCAGAACGATGGATGGATAACT; F2:AAA CAGTTATCCATCCATCGTTCT. Plasmid of pHR-SFFV-dCas9-BFP-KRAB was a gift from Liang lab (46911, Addgene). SgRNAs fragments were cloned into lentiguide-puro (52963, Addgene). Successfully knock-down clones were enriched through FACS to obtain eGFP-BFP positive cells.

## Cell proliferation

Proliferation of metastatic cells and UPR genes knockdown cells were measured using the CCK8 Assay (C6005, New Cell & Molecular Biotech, China). After 24, 48, 72, and 96 h of cell culture, cells were incubated with the CCK8 dye (10% CCK8 in M15 medium) for 3 h at 37 °C. Absorbance values at 450 nm were examined by a microplate reader (Bio-TEK Instrument, USA).

## Tail vein injection of CTC-like cells and metastatic cells

Ts6 + 8 (1#) CTC-like cells and metastatic cells (Ts15-Met and Ts8 + 15-Met, as well as UPR genes knockdown clones) were dissociated with 0.05% trypsin-EDTA and sedimented for 30 min to remove iMEFs, followed by tail vein injection of $5 \times 10^5$ CTC-like cells or $2 \times 10^5$ metastatic cells into SCID/Beige mice (5 mice, male, 7 weeks of age). Approximately 4–5 weeks after the injection, mice were observed by IVIS to observe the metastatic signal distribution.

## Western blot

Samples were lysed in RIPA solution. Protein content was quantified using the BCA method (Thermo Fisher). Proteins were loaded onto a 12% SDS-polyacrylamide gel and blotted onto a polyvinylidene fluoride (PVDF) membrane (Millipore). The blotted PVDF membranes were blocked with 5% milk or 3% BSA in Tris-buffered saline with Tween 20 and incubated overnight at 4 °C with the primary antibody. Proteins were finally detected with a horseradish peroxidase (HRP)-conjugated secondary antibody (CST, 7074).

## Total proteasome activity test

Cells were harvested and lysed in lysis buffer (50 mM Tris, pH 7.5, 150 mM NaCl, 1% glycerol, 5 mM $MgCl_2$, ATP, cOmplete Protease Inhibitor Cocktail, PMSF). The protein concentration was determined using a BCA Protein Assay kit (Pierce). The chymotryptic activity of the proteasome was assayed via the hydrolysis of the fluorogenic peptide succinyl-Leu-Leu-Val-Tyr-7-amino-4-methylcoumarin (suc-LLVY-AMC) (Bachem). Approximately 10–30 μg of protein from each extract was incubated with 200 mM suc-LLVY-AMC in 50 mM Tris (pH 7.8) and 1 mM DTT in a total volume of 100 μL. Fluorescence was read on a spectrofluorometer using 390-nm excitation and 460-nm emission filters.

## Detection of carbonylated proteins by Western blot analysis

Samples used to identify carbonylated proteins were prepared according to the OxyBlot™ Protein Oxidation Detection Kit protocol (Catalog No. S7150). Briefly, a protein solution was prepared by adding 1–2% 2-mercaptoethanol (as a reducing agent) to the lysis buffer to prevent the oxidation of proteins, which may occur after cell lysis. Proteins were denatured by adding SDS at a final concentration of 6%. Carbonyl groups were derived by adding DNPH solution. Samples were incubated at room temperature for 15 min. Neutralization solution was then added to stop the reaction. The treated samples were subsequently loaded onto a polyacrylamide gel for Western blot analysis. The primary antibody was diluted to 1:150 with blocking/dilution buffer. The secondary antibody was diluted at 1:300.

## Drugs and treatments

Oleuropein was purchased from MedChemExpress (HY-N0292) and was dissolved in 0.9% NaCl at the concentration of 5 mg/mL before use. Briefly, in vitro, Oleuropein (0.5 μg/mL) was added during the first three days of EB differentiation. In vivo, Oleuropein (20 mg/kg/day) was intraperitoneally injected every other day when teratomas became palpable. 4-PBA (Sigma, P21005-100G) was dissolved in 1 N NaOH followed by 0.9% NaCl at the concentration of 50 mg/mL, and then was intraperitoneally injected (300 mg/kg/day) every other day when tumor sizes reached 50–100 mm³. Both Oleuropein and 4-PBA administration lasted for the whole observation period.

## Antibodies

Antibodies were used at the following dilutions: rabbit polyclonal anti-ATF6, 1:500 (Abcam, catalog number ab37149); mouse monoclonal anti-BIP, 1:1000 (BD, catalog number 610978); rabbit monoclonal anti-p-PERK, 1:1000 (Cell Signaling Technology, catalog number 3179); rabbit polyclonal anti-p-EIF2A, 1:1000 (Cell Signaling Technology, catalog number 9721); rabbit monoclonal anti-EIF2A, 1:1000 (Abcam, catalog number ab169528), rabbit polyclonal anti-PA28α, 1:1000 (ENZO, catalog number BML-PW8185); rabbit polyclonal anti-β5i, 1:1000 (ENZO, catalog number BML-PW8400); goat polyclonal anti-ACTIN, 1:1000 (Santa Cruz Biotechnology, catalog number sc-1616); mouse monoclonal anti-OCT3/4, 1:1000 (Santa Cruz Biotechnology, catalog number sc-5279); rabbit monoclonal anti-GFP, 1:1000 (Cell Signaling Technology, catalog number 2956); Mouse monoclonal Anti-VINCULIN, 1:10,000 (Sigma, catalog number V9131); rabbit polyclonal anti-XBP1, 1:1000 (Abcam, catalog number ab37152).

## Preparation of DNA for sequencing

The samples were processed using a QIAamp DNA Mini Kit (Qiagen, Germany, Catalog no. 51304) according to the manufacturer's protocol. In brief, samples of approximately 25 mg of frozen tissue or 10⁶ cells were lysed in AL solution containing proteinase K, after which ethanol was added, and the mixture was purified with a QIAamp Mini spin column, buffer AW1 and buffer AW2. The purified DNA was dissolved in 20 μL DEPC-treated water and stored at −20 °C for future use.

## Whole-genome library preparation and sequencing

The whole-genome libraries were constructed with the NEBNext® Ultra™ II DNA Library Prep Kit for Illumina® (NEB, Catalog no. E7645L) according to the manufacturer's protocol. In brief, approximately 200 ng genomic DNA was randomly sheared into fragments of less than 300 bp by using a Covaris system (Covaris). The fragmented DNA was then converted to repaired DNA with 5′ phosphorylated, 3′ dA-tailed ends. Then, adapters were added to both ends of the repaired DNA. Agencourt AMPure XP SPRI beads (Beckman Coulter, USA) were used for the size selection of adapter-ligated DNA. The purified product was subsequently amplified with P5 and P7 primers carrying sequences. The PCR products were cleaned up, and the final library was sequenced with a NovaSeq 6000 System. Paired-end (PE) reads (2 × 150 bp) were generated.

## Whole-exome library preparation and sequencing

The exome-captured libraries were generated using the SureSelectXT Mouse All Exon Kit (Agilent Technologies, Catalog no. 5190-4642) according to the manufacturer's protocol. Briefly, the whole-genome library was built via a similar approach to that described above. The exon regions of the library were captured, amplified, and purified. The final library was built, quality checked, and sequenced with a NovaSeq 6000 System to generate 2 × 150 bp PE reads.

## Quality control and preprocessing of FASTQ files

Quality control, quality filtering, adapter trimming, and per-read quality pruning were performed using fastp (version: 0.20.0)[55]. Some parameters were adjusted. Only bases with a Phred quality higher than 19 were qualified. In each read, a maximum of 50% of bases are allowed to be unqualified. Reads shorter than 36 were discarded. PolyX trimming of the 3′ ends was enabled.

## Analysis of WES data for insertion/deletion (INDEL) and single nucleotide variant (SNV) calling

Filtered reads generated from WES sequencing were aligned to the *Mus musculus* reference genome GRCm38 (by bwa (version: 0.7.17-r1188))[56]. The products were sorted with SAMtools (version: 1.7)[57] and then processed with the GATK toolkit (version: 3.8-0-ge9d806836)[58,59]. Briefly, the duplicated reads were marked and removed with *MarkDuplicates*, and the accuracy of each base call was estimated with *BaseRecalibrator*. INDELs and SNVs were called using *MuTect2*. WES analysis was carried out on primary and metastatic tumors as well as normal tissues from the same mice. By comparing the primary tumor with the paired normal tissue, INDELs and SNVs belonging to the germline background were obtained. Then, we compared the metastatic tumor with the paired primary tumor using the default parameters and preserved variants tagged with "PASS". Variants that had been identified as germline background and had been registered in the dbSNP database were discarded as well. Information on mutations stored in variant call format (VCF) files was annotated using vep (version: 101.0)[60] and converted to mutation annotation format (MAF) files using vcf2maf (version: 1.6.21). The MAF files were summarized, analyzed, annotated, and visualized with maftools (version: 2.6.05) with the default parameters[61].

## Copy number analysis based on WGS

Filtered reads obtained from WGS were aligned and sorted in the same way as in WES. The target BAM files were processed together with the WT ESCs BAM file to generate seqz files using sequenza-utils (version: 3.0.0)[62]. The window size used for binning the original seqz file was set to 200. Then, we used the Sequenza R package (version: 3.2.0) to obtain genome-wide integer copy numbers, except for the Y chromosome.

## Single-cell transcriptome library preparation and sequencing

Fresh tissue samples were stored in MACS BSA Stock Solution (Miltenyi Biotec, Catalog no. 130-091-376) at 4 °C and were processed within 2 h. Single-cell suspensions were obtained using the Tumor Dissociation Kit (Miltenyi Biotec, Catalog no. 30-096-730) with a Gentle MACS Octo Dissociator (Miltenyi Biotec, USA) according to the manufacturer's protocol. Single-cell 3′ transcriptome libraries for each sample were constructed using the Single Cell 3′ Library and Gel Bead Kit V3 (10x Genomics, Catalog no. 1000075) and the Chromium Single Cell B Chip Kit (10x Genomics, Catalog no. 1000074) following the manufacturer's protocol. In short, approximately 6000–10,000 single cells were loaded onto a microfluidic chip along with gel beads containing barcoded oligonucleotides, reverse transcription (RT) reagents, and partitioning oil. Nanoliter-scale reaction vesicles were formed using microfluidic partitioning. Within each vesicle, a single cell was lysed, and its mRNAs were released, captured, and reverse transcribed to generate cDNAs containing cell-specific barcodes. The barcoded cDNAs were then converted into a next-generation sequencing (NGS) library in a bulk reaction. The NovaSeq 6000 System was utilized to sequence the NGS library.

## Preprocessing of single-cell transcriptome sequencing (scRNA-seq) data

Cell Ranger (version: 3.1.0) was used to build a reference and preprocess scRNA-seq data. To capture the expression information of enhanced green fluorescent protein (EGFP) and luciferase, a custom reference was built following the instructions of 10x Genomics (https://support.10xgenomics.com/single-cell-gene-expression/software/pipelines/latest/using/t-utorial_mr#marker). In brief, the reference genome and gene transfer format (GTF)

files of *Mus musculus* were downloaded from Ensembl (version 102). The *Mus musculus* GFT file was filtered by using the *mkgtf* module of Cell Ranger. Then, the EGFP and luciferase information was converted into FASTA format and GTF and was added to the tails of corresponding *Mus musculus* files. The *mkref* module of Cell Ranger was run to create a ready-to-use reference.

The *Count* module of Cell Ranger was used to perform alignment, filtering, barcode counting, and unique molecular identification (UMI) counting. Then, the count matrices were generated for each sample. Seurat (version: 4.0.1) was used to process the count matrices[63]. Briefly, cells with fewer than 200 detected genes or mitochondrial transcript ratios higher than 25% were discarded. For each sample, we used the *SCTransform* function to scale and normalize the expression matrix and extract highly variable genes (HVGs). Then, the matrices and HVGs were merged to build a merged object.

### Dimensionality reduction and annotation of major cell clusters based on scRNA-seq data
Dimension reduction and clustering were applied to identify major cell clusters. We first performed principal component analysis (PCA) of the normalized matrix. Thirty principal components (PCs) were used in Seurat's *FindNeighbors* function to construct a KNN graph and refine the edge weights between each pair of cells. We next applied Seurat's *FindClusters* function, which is based on the Louvain algorithm, to cluster the cells. The *resolution* parameter of *FindClusters* was set to 0.8. The result of cell clustering was visualized in a UMAP plot. For UMAP reduction, 30 PCs were used, *min.dist* was set to 0.8 and *spread* was set to 1.5. Marker genes of each cell cluster were identified using Seurat's *FindAllMarkers* function. Genes with an adjusted *p* value lower than 0.05 and log$_2$(fold-change) greater than 0.5 were taken as marker genes. *DppaSa*$^+$ or *EGFP*$^+$ clusters or clusters with significant CNVs were identified as cells originating from embryonic stem cells and aneuploid embryonic stem cells (ES). Canonical markers were used to identify the remaining clusters (Supplementary Table 3). ES cells expressing *Col3a1* and *Col1a1* were identified as fibroblast-like ES (ES_FB) cells. ES cells expressing *Nes* and *Msi1* were identified as neural-stem-like ES (ES_NSC) cells. ES cells expressing *Gfap* were identified as Schwann cell-like ES (ES_Schw) cells. ES cells expressing *Map2* and *Tubb3* were identified as neural cell-like ES (ES_NC) cells. ES cells expressing *Tnnc2, Myl1* and *Des* were identified as muscle cell-like ES (ES_Musc) cells. ES_Ori cells mainly originated from cell line samples and high expressing *DppaSa*. Another group of *DppaSa*$^+$ ES cells composed of cells from tissue samples were identified as stem-like ES (ES_Stem) cells.

### Gene ontology (GO) analysis
Metascape was used to perform GO analysis with the default parameters. Genes with an adjusted *p* value lower than 0.05 and an absolute log$_2$(fold-change) value greater than 0.5 were considered as DEGs[64]. Up- and down-regulated DEGs were calculated separately.

### Visualizing large-scale copy number variations in scRNA-seq data
Large-scale chromosomal copy number variations at the single-cell level were detected by using inferCNV (version: 1.6.0, inferCNV of the Trinity CTAT Project. https://github.com/broadinstitute/inferCNV). To validate the annotation accuracy, cells other than ES cells were combined to build a reference set. Then, the expression intensities of genes along the genome were calculated. The relative expression intensities across each chromosome are illustrated in a heatmap, indicating which genome regions were overabundant or less abundant. Specifically, the *CreateInfercnvObject* function was used to build inferCNV objects. Cells belonging to the granulocyte, macrophage, DC, ILC, erythroid, endothelium, fibroblast, and epithelium groups were annotated as 'normal' reference cells. The *run* function

was then employed to infer large-scale chromosomal copy number variations. The *cutoff* parameter was set to 0.1 to match the sparse matrix generated by 10x Genomics. The denoise and HMM settings were set to FALSE. To visualize the copy number variations across ES cells, WT ES cells were used as a reference.

### Cell trajectory analysis
We used the R package Monocle3 (version: 1.0.0) to reconstruct the differentiation trajectory of ES cells[65–68]. The Monocle object was built using the raw count matrix and cluster annotation derived from the Seurat object. The *preprocess_cds* function was used to normalize the matrix. The *align_cds* function was used to remove batch effects. The *reduce_dimension* function was used to reduce the dimensions, during which *reduction_method* was set to UMAP, *umap.min_dist* was set to 0.4, and *umap.n_neighbors* was set to 50. The trajectory was constructed by using the *learn_graph* function. Then, the pseudotime was predicted using the *order_cell* function. To extract genes that were differentially expressed along the trajectory, the *graph_test* function was used with the *neighbor_graph* parameter set as principal_graph. Genes with q-value equal to 0 and Moran's I value larger than 0.2 were taken as a subset and then clustered into coregulated gene modules by using the *find_gene_modules* function.

### Bulk RNA-seq
Total RNA was extracted from harvested EBs using TRIzol reagent (Invitrogen) following standard manufacturer's instructions. 1 µg of total RNA was randomly fragmented and reverse-transcribed using the PrimeScript II 1st strand cDNA Synthesis Kit (TaKaRa, D6210A) according to the manufacturer's instructions. VAHTS® Universal V8 RNA-seq Library Prep Kit for Illumina was used to prepare library for sequencing, about 60 million reads raw data were generated per sample from Novaseq 6000 PE150 equipment. Quality control, quality filtering, adapter trimming, and per-read quality pruning were performed using fastp[55] (version: 0.23.2) and MultiQC[69] (v 1.10.1). Filtered reads generated from RNA sequencing were aligned to the Mus musculus reference by STAR[70] (v 2.7.4a). The "--quant-Mode" was set as *GeneCounts* to generate the count matrices. DESeq2[71] (v1.30.1) were used to generate DEGs. Genes with an adjusted *p* value lower than 0.05 and absolute value of log$_2$(fold-change) greater than 0.5 were taken as DEGs. Seurat (version: 4.0.1) was used to evaluate the scores of gene sets. Of which, the *CreateSeuratObject* function was used to create objects, and we then used the *SCTransform* function to scale and normalize the expression matrix. The *AddModuleScore* was used to calculate the score of specific gene sets.

### Statistics and reproducibility
The data are shown as the means ± SD. Significant differences between two groups were analyzed with Student's *t*-test. *p* < 0.05 was considered statistically significant, as indicated by asterisks (*$p$ < 0.05, **$p$ < 0.01, ***$p$ < 0.001). PCR results and Histologic analysis including IHC and HE staining are reproducibility available. Micrograph photos of cell lines used in this work are reproducibility available.

### Reporting summary
Further information on research design is available in the Nature Portfolio Reporting Summary linked to this article.

## Data availability
The raw sequencing data generated in this study have been deposited in NCBI (PRJNA790979), and are publicly available as of the date of publication. Result files and intermediate files for bioinformatics analysis are available at https://doi.org/10.5281/zenodo.10369215. Source data are provided with this paper.

## Code availability

The code involved in this work is available at https://doi.org/10.5281/zenodo.10320329.

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

## Acknowledgements

We thank R. Zhang for assistance with scRNA-seq analysis, H.W. Wu from Peking Union Medical College Hospital for performing pathological examinations, X.B. Qiu from Beijing Normal University for kindly providing materials for the proteasome viability assay, and H.B. Zhang and J.B. Liang from Peking Union Medical College for kindly providing plasmids for lentivirus packaging. This work was financially supported by the CAMS Innovation Fund for Medical Sciences (2021-I2M-1-019), the National Key Research and Development Program of China (2019YFC1315702, 2022YFC2504602, 2023YFC2705801), the National Natural Science Foundation of China (32170771, 31970813, 92042303, 82241230, T2125002, 82341007, 92259303), and the State Key Laboratory Special Fund (2060204). This work has been supported by the New Cornerstone Science Foundation through the XPLORER PRIZE.

## Author contributions

Y.H. and F.B. conceived the study. Y.H., F.B., R.X., and M.Z. designed the experiments. R.X. and M.Z. performed most of the experiments with the help of S.D., M.L., and T.Z. D.X., Z.C., and R.L. performed the data analysis. L.C. performed the proteasome experiments. Y.H., F.B., R.X., M.Z., and D.X. wrote the manuscript.

## Competing interests

The authors declare no competing interests.
