## [Peer Review File · Nature Communications]

REVIEWER COMMENTS

Reviewer #1 (Remarks to the Author):

In this manuscript, Xiao, Xu, Zhang, and Chen et al. used that teratoma model to study the role of aneuploidy in metastasis and showed that, while WT ES cells-derived teratoma is nearly non-metastatic upon transplantation, trisomic ES cells-derived teratoma possess metastatic capability. They performed genomic analysis of primary and metastatic lesions and failed to find any changes related to oncogene activation or tumor suppressor genes inactivation, alluding that aneuploidy likely directly promotes metastasis. To corroborate this hypothesis, the authors reversed the aneuploid state back to diploid in these cells and this results in suppression of metastasis. In order to investigate the mechanism behind aneuploidy induced metastasis, they performed and analyzed single cell transcriptome of trisomic and diploid teratoma, where they found that trisomic teratoma possesses stronger stem cell traits, consistent with their previous published findings ((PMID: 27558554). Furthermore, they demonstrated that metastatic trisomic teratoma exhibited less proteasome activity yet a higher level of unfolded protein response (UPR). Forced activation of proteasome activity or inhibition of UPR by small molecule inhibitors suppress the metastatic activity of aneuploid ES cells.

Aneuploidy is a state of chromosomes with numerical and/or structural deviation and is a ubiquitous feature of cancer cells. In addition to being a source of tumor heterogeneity, aneuploidy can either promote or hinder tumor progression in a cell-type and time-dependent manner, known as aneuploidy paradox. The lab of Dr. Huang has previously developed a series of trisomic mouse embryonic stem (ES) cell lines to evaluate the role of aneuploidy in teratoma (PMID: 27558554). They showed that trisomic ES cells possess lower differentiation ability and higher teratoma formation ability than their diploid counterparts.

We think this study is highly relevant given the high prevalence of aneuploid tumors and uses an elegant model previously developed by the same group. The key observations are in line with existing literature and are interesting to the aneuploidy field. It adds significantly to the field. However, we think there are still number of concerns we would like to express and should be addressed before this paper should be published.

1) By showing that reversing the aneuploidy in stem cell can suppress metastasis, the authors concluded that aneuploidy itself is the contributing factor to elevated metastasis potential while concomitantly ruling out that genetic alterations related to oncogenes/tumor suppressor due to aneuploidy state is the driving force behind the metastasis potential. However, we think it is not justified to draw such a strong conclusion with negative data. The authors should perform additional experiments to make this point.

2) In their single cell analysis, they included in vitro cultured cells isolated from in vivo. Since the circumstances in vivo is quite different from in vitro conditions., it is not necessary to include cells in vitro into the scRNA analysis.

- 3) The authors showed that by applying proteasome activators or UPR inhibitors on the trisomic cells suppress their metastasis. Given that these drugs often have off-target effects, the authors should exclude this possibility while also showing the on-target inhibition or activation of these drugs (maybe through KD or KO of UPR genes?)
- 4) In Fig6j, proper vehicle control should be done at the same time.
- 5) Number of literatures have demonstrated importance of UPR response in immune landscape of metastatic lesion. It might be interesting to test if this is the case for the aneuploid ES cells as well
- 6) Finally it is critical to distinguish between aneuploidy and chromosomal instability. While the authors make the claim that it is aneuploidy itself that drives metastasis there is prior work suggesting that the ongoing instability can also drive metastasis. While the two are related and linked they are not synonymous. The authors should address this point experimentally.

Reviewer #2 (Remarks to the Author):

Aneuploid embryonic stem cells drive teratoma metastasis

“Aneuploid embryonic stem cells drive teratoma metastasis” manuscript shows how several chromosome gains (trisomies), in the context of stable CIN, promote teratoma invasiveness and metastasis. This is a novel and interesting finding, in the current discussion whether aneuploidy can affect tumor progression by itself without requiring/inducing chromosome instability. However, the authors should address major and minor comments prior acceptance of the manuscript:

Major comments:

- 1- Seems that a lot of effect is driven by the gain of chromosome 8.
 - a. Figure 1: The % of metastasis developed ranges from 31% to 63% depending of the background, but the highest rates are consistently found in the context of chromosome 8 trisomy (Ts8 (62.5%) Ts6+8 (45%) Ts8+15 (62.9%)).
 - b. Figure 2 & Extended Figure 4: WGS revealed one Ts6 changes to Ts6+8 and one Ts8 changes to Ts6+8
 - c. Figure 4: CTCs were only successfully isolated from Ts6+8
 - d. Figure 6a: Proteasome activity reduction significant only in Ts6 and Ts8
 - e. Mice Chromosome 8 contains 75% of the genes encoding for rRNAs.
(<https://www.ncbi.nlm.nih.gov/genome?term=txid10091%5BOrganism%3AAnoexp%5D&cmd=DetailsSear>)

ch) This might be also affecting the results of the pathways affected in extended figure 7d, since samples carrying trisomy 8 are one of the main contributors to this analysis (ES_Stem vs ES_Ori)

All this data should be mentioned in the manuscript and included in the discussion with rationale about possible explanations for this phenomenon.

2- ScRNAseq comments:

a. From figure 5c and extended figure 7c, it seems that the contribution to the ES_Stem population is highly enriched with cells from Ts11_M. In the data from InferCNV, gain of chr11 is the only clearly detectable trisomy traced in this cell population and it's present in almost 50% of the cases. Also the ES_Stem population existing in the WT samples represents only 1% of the total cells on the samples. So the comparison in figure 5f comparing differentially expressed genes between ES_Ori or ES_Stem between WT and AC might be biased by the enrichment of specific genotypes.

b. Extended figure 7b seems to show that metastatic samples are enriched in cells from the immune compartment coming from the mice (DCs and Macrophages). This might be worth some discussion.

c. Figure 5e: The authors in line 213 quote "we noted that cancer-related genes such as S100 family members (S100a8, S100a9, S100g) 28,29 and Hbb-bs 30 were upregulated". According to the figure these genes are apparently upregulated in intermediate timepoints but not at the latest timepoint. Additionally, there is no information provided about the other clusters in the figure, so the authors should describe these clusters or remove them from the figure.

d. The authors do not indicate whether the primary and metastases samples are paired. And the data from extended table 1 and 2 does not clarify this fact. The extended data tables 1 and 2 should be indicate which samples are primary teratomas and which ones are metastasis.

3- Proteasome activity and differentiation:

a. In line 239 authors quote "Unlike WT cells, the total proteasome activity in aneuploid cells quickly declined during the early stage of embryoid body (EB) formation (Fig. 6a)". However, in figure 6a at the D6 timepoint, the WT proteasome activity has declined at a similar level of 50% of the aneuploid samples. This might be related to specific trisomies rather than a general aneuploidy/trisomy effect.

b. I would like to see for figures 6d and 6f a foldchange graph between timepoint D0 and D6 or D2 and D4 if there is only data available from these last two timepoints.

c. In line 254 quote "Oleuropein effectively reduced aneuploid teratoma volume and partially rescued the deficiency of aneuploid ESC differentiation, as evidenced by significantly decreased proportions of Oct4+ cells in the teratoma". Figure 6h actually shows WT oct4 expression under treatment but not vehicle. It seems that the Oleuperin effect is more related to contain the OCT4+ cells in dense clusters hampering their spread, but they remain highly positive for OCT4. Maybe this is affecting the invasiveness potential of OCT4+ cells. I suggest testing in vitro invasion, migration, and proliferation under treatment.

d. Additionally, the authors do not provide in vitro evidence showing that Oleuropein rescues proteasome activity in the ESC cell lines used prior to in vivo treatment.

e. In line 252 quote “These results suggest that insufficiency of proteasome activity and overactivated UPR might underlie the differentiation defects of aneuploid cells.” I don’t think the authors actually showed data regarding differentiation, but rather stemness maintenance. To support this would be great to have Histology, IHC, gene expression data and/or protein levels on WT and AC teratoma samples.

f. Bulk RNAseq in differentiated EBs from WT or trisomic cell will help to elucidate the effect of these gains in differentiation and tumor invasion and metastasis.

Minor Comments

4- CTCs experiment

a. In my opinion, the most interesting question in this experiment is missing. Finding CTCs when there is already metastasis present it is expected. Would have been more interesting to withdraw blood before removing the primary teratoma to avoid any possible bias introduced during the surgery.

b. Extended figure 6b. The authors showed positivity for OCT4 but they do not show that the cells retained in vitro eGFP.

The authors should address any possible limitations of the CTCs study in the discussion including why CTCs were only successfully isolated from Ts6+8.

5- Trisomy correction: the image showing the phenotype change from Ts11 to Di11 (figure 3b) doesn’t look really different. Authors should provide some quantitative assessment/measurement to prove this or select a better image.

6- The manuscript talks about aneuploidy in general when it is only focused on gain of specific chromosomes (Trisomies). This is relevant because aneuploidy also includes chromosome deletion. I would suggest to change the manuscript title from Aneuploidy to Trisomy.

7- The discussion section need to be more elaborated, commenting on possible caveats and limitations of the presented study and comparing the results to previous findings from other groups.

Point-by-point response

Manuscript ID: NCOMMS-23-04507-T

We appreciate the valuable comments from the reviewers, which greatly helped us to improve the manuscript. Please see our point-by-point responses below.

Response to Reviewer #1

Reviewer #1 (Remarks to the Author):

In this manuscript, Xiao, Xu, Zhang, and Chen et al. used that teratoma model to study the role of aneuploidy in metastasis and showed that, while WT ES cells-derived teratoma is nearly non-metastatic upon transplantation, trisomic ES cells-derived teratoma possess metastatic capability. They performed genomic analysis of primary and metastatic lesions and failed to find any changes related to oncogene activation or tumor suppressor genes inactivation, alluding that aneuploidy likely directly promotes metastasis. To corroborate this hypothesis, the authors reversed the aneuploid state back to diploid in these cells and this results in suppression of metastasis. In order to investigate the mechanism behind aneuploidy induced metastasis, they performed and analyzed single cell transcriptome of trisomic and diploid teratoma, where they found that trisomic teratoma possesses stronger stem cell traits, consistent with their previous published findings ((PMID: 27558554). Furthermore, they demonstrated that metastatic trisomic teratoma exhibited less proteasome activity yet a higher level of unfolded protein response (UPR). Forced activation of proteasome activity or inhibition of UPR by small molecule inhibitors suppress the metastatic activity of aneuploid ES cells.

Aneuploidy is a state of chromosomes with numerical and/or structural deviation and is a ubiquitous feature of cancer cells. In addition to being a source of tumor heterogeneity, aneuploidy can either promote or hinder tumor progression in a cell-type and time-dependent manner, known as aneuploidy paradox. The lab of Dr. Huang has previously developed a series of trisomic mouse embryonic stem (ES) cell lines to evaluate the role of aneuploidy in teratoma (PMID: 27558554). They showed that trisomic ES cells possess lower differentiation ability and higher teratoma formation ability than their diploid counterparts.

We think this study is highly relevant given the high prevalence of aneuploid tumors and uses an elegant model previously developed by the same group. The key observations are in line with existing literature and are interesting to the aneuploidy field. It adds significantly to the field. However, we think there are still number of concerns we would like to express and should be addressed before this paper should be published.

1) By showing that reversing the aneuploidy in stem cell can suppress metastasis, the authors

concluded that aneuploidy itself is the contributing factor to elevated metastasis potential while concomitantly ruling out that genetic alterations related to oncogenes/tumor suppressor due to aneuploidy state is the driving force behind the metastasis potential. However, we think it is not justified to draw such a strong conclusion with negative data. The authors should perform additional experiments to make this point.

Response: We thank the reviewer for the comment. Although we all know that aneuploidy is a ubiquitous feature of cancer, approximately 90% of solid tumors and 70% of hematopoietic malignancies display a certain degree of aneuploidy, whether aneuploidy is a driving force of cancer initiation and progression is a long-lasting debate.

The strategy of trisomy correction was first introduced by *Li et al.* to generate disomic iPSCs from Down syndrome (DS) patients (*Li et al.*, 2012). A *TKNEO* fusion gene encoding thymidine kinase was introduced into chromosome 21 of DS iPSCs. This allows a negative ganciclovir (GCV) selection for cells that have lost chromosomes harboring *TKNEO* gene. We then used a similar strategy to select for ESCs with spontaneous trisomic chromosome loss, and established a causal relationship between aneuploidy and enhanced teratoma formation (*Zhang et al.*, 2016). Moreover, a recently published work reported the elimination of specific aneuploidies from established cancer cell lines by using ReDACT (Restoring Disomy in Aneuploid cells using CRISPR Targeting), and the removal of extra chromosomes compromise cancer malignancy (*Girish et al.*, 2023). By using this strategy, they found recurrent aneuploidies could represent a type of “cancer addiction”. Here, we used the similar strategy of trisomy correction as we reported before, and found the removal of trisomic chromosomes could compromise teratoma metastasis. We thus point out that aneuploidy itself might be one of driving forces of metastasis based on our cell model and studies.

However, we did not exclude the critical roles of oncogenes (OGs) and tumor suppressor genes (TSGs) brought by aneuploidy in cancer malignancy. Gene expression levels often scale with gene/chromosome copy number (*Pavelka et al.*, 2010). It was reported that OGs are prone to accumulate in chromosome gain regions, while TSGs tend to accumulate in chromosome loss regions (*Davoli et al.*, 2013). Accordingly, the genes located on trisomic chromosomes are largely overexpressed in our aneuploid mESCs (**Response figure 1 and 2**). Among them, the well-known OG *C-myc* is located on mouse chromosome 15, and the remarkable TSG *Trp53* is located on mouse chromosome 11. Noticeably, Ts11 teratoma bearing mice showed obvious lung metastasis, albeit the P53 expression was 1.77 fold-increase in Ts11 compared with WT mESCs. On the other hand, the amplification of OGs brought by aneuploidy might facilitate tumor malignancy as well. More importantly, one trisomic chromosome harbors numerous OGs and TSGs simultaneously, it is quite difficult to attribute the aneuploid ESC-related teratoma metastases to the overexpression of a few particular OGs. At the current stage, we cannot accurately calculate the effects brought by the expression changes of many OGs and TSGs while reversing aneuploidy to diploid state. As to this, we would like to conclude that

aneuploidy is one of the contributing factors to teratoma metastasis (Pages 7, lines 129-130).

Response figure 1. Western blot analysis of trisomy-specific proteins expression in wild-type (WT), and trisomic ESCs. K-ras, Nanog, Lrp6 and Gapdh at chromosome 6; Herpud1, Hmox1, Idhd and Ufsp2 at chromosome 8; Erlec1, P53, C53 and Gaa at chromosome 11; C-myc, Sox10, Rgs22 and α -tubulin at chromosome 15. The values are related to those of WT ESCs, which were assigned a value of 1.0.

	Gene number	Ts6_up	Ts6_down	Ts8_up	Ts8_down	Ts11_up	Ts11_down	Ts15_up	Ts15_down
chr1	1352	5.1%	6.0%	4.2%	5.6%	4.4%	4.4%	4.8%	3.1%
chr2	2055	3.2%	4.9%	2.7%	4.3%	3.6%	4.2%	3.3%	2.7%
chr3	1145	5.9%	5.4%	3.9%	6.3%	4.8%	5.2%	4.9%	3.2%
chr4	1456	5.2%	5.4%	3.1%	5.0%	4.3%	4.9%	4.5%	3.9%
chr5	1382	4.6%	5.2%	3.5%	6.2%	4.3%	5.7%	4.8%	4.0%
chr6	1293	29.5%	1.5%	2.8%	5.6%	4.3%	5.2%	4.1%	4.3%
chr7	2201	3.9%	5.0%	2.0%	5.0%	2.5%	6.1%	3.3%	3.3%
chr8	1174	5.0%	5.0%	41.7%	1.8%	3.7%	6.7%	4.3%	4.3%
chr9	1358	5.9%	3.4%	4.0%	5.2%	3.8%	4.7%	4.2%	3.6%
chr10	1151	4.1%	5.2%	3.0%	5.8%	3.6%	4.6%	4.4%	2.9%
chr11	1837	5.6%	3.8%	3.4%	5.8%	37.2%	0.4%	3.9%	3.5%
chr12	833	4.3%	7.6%	5.0%	4.6%	4.2%	5.8%	3.5%	3.1%
chr13	936	2.9%	4.3%	4.0%	2.9%	4.1%	5.6%	2.5%	1.9%
chr14	879	4.3%	5.8%	3.9%	5.2%	3.9%	5.1%	3.9%	3.0%
chr15	907	4.5%	6.5%	4.1%	6.9%	5.2%	4.7%	26.2%	0.3%
chr16	754	4.5%	5.4%	2.9%	6.1%	4.2%	4.1%	4.8%	2.0%
chr17	1194	4.9%	5.0%	2.4%	5.9%	3.1%	5.4%	2.9%	4.1%
chr18	600	3.8%	5.8%	3.3%	4.5%	4.7%	4.8%	4.3%	3.0%
chr19	791	5.6%	5.8%	3.4%	5.1%	5.3%	4.4%	3.5%	3.3%
chrX	1063	4.3%	5.9%	3.1%	5.4%	5.3%	5.4%	4.7%	3.2%

Response figure 2. Summary of the percentage of both up- and down-regulated genes per chromosome.

REFERENCE

- Li, L. B. et al. Trisomy correction in Down syndrome induced pluripotent stem cells. *Cell Stem Cell* (2012) 11, 615-619.
- Zhang, M. L. et al. Aneuploid embryonic stem cells exhibit impaired differentiation and increased neoplastic potential. *EMBO J* (2016) 35: 2285-2300.
- Girish, V. et al. Oncogene-like addiction to aneuploidy in human cancers. *Science* (2023) 381: eadg4521.
- Pavelka, N. et al. Aneuploidy confers quantitative proteome changes and phenotypic variation in budding yeast. *Nature* (2010) 468: 321-325.
- Davoli, T. et al. Cumulative haploinsufficiency and triplosensitivity drive aneuploidy patterns and shape the cancer genome. *Cell* (2013) 155: 948-962.

2) In their single cell analysis, they included in vitro cultured cells isolated from in vivo. Since the circumstances in vivo is quite different from in vitro conditions., it is not necessary to include cells in vitro into the scRNA analysis.

Response: We thank the reviewer for the comment. In fact, we did not include the *in vitro* cultured cells isolated from *in vivo* teratoma samples in the single cell analysis. There are mainly two sources: one is the cells isolated from primary teratomas and metastases, and the other is the original ESC lines cultured *in vitro*. We need the original ESC lines cultured *in vitro* as an “anchor” to predict the cell differentiation trajectory. To evaluate whether the introduction of *in vitro* cultured ES cell lines would affect the clustering of cells isolated from tissues, we removed the *in vitro* data and performed re-clustering and annotation. It showed that the cell annotation results highly overlapped with the original cell annotation results (**Response figure 3**). This indicates that the introduction of *in vitro* cultured ESC line data does not affect our primary conclusions of the single-cell data of aneuploid tumors.

Response figure 3. Heatmap displaying the consistency of cell annotation before and after removing the *in vitro* cultured cell lines.

3) The authors showed that by applying proteasome activators or UPR inhibitors on the trisomic cells suppress their metastasis. Given that these drugs often have off-target effects, the authors should exclude this possibility while also showing the on-target inhibition or activation of these drugs (maybe through KD or KO of UPR genes?)

Response: We appreciate the reviewer's suggestions. To confirm the role of UPR inhibition in metastasis, we performed knockdown of three critical UPR genes, *Atf6*, *Eif2a* and *Xbp1*, by CRISPRi (*Gilbert et al.*, 2014) in aneuploid metastatic cells isolated and cultured from metastatic mice of Ts15 (Ts15-Met) and Ts8+15 (Ts8+15-Met). These cells were then injected into SCID mice via tail vein injection. Here, considering the long timeline of metastasis observation after subcutaneous injection, we performed mouse tail vein injection of the cultured metastatic cells, which is a faster and more accurate way to measure the malignancy of aneuploid cells, like we did for CTCs in our study. We found that knockdown of UPR genes significantly reduced the metastatic efficiency of Ts15-Met and Ts8+15-Met. The number of lung metastatic nodules decreased significantly after UPR genes knockdown. Therefore, similar with the result of UPR inhibitor drugs, targeting UPR genes in aneuploid metastatic cells can help mitigate metastasis driven by aneuploidy (**Response figure 4**). We have updated and added these results to Figure 6 and Supplementary Fig. 14 in the revised manuscript.

Response figure 4. UPR genes knock down in metastatic cell lines showed decreased metastatic efficiency. (a) CRISPRi plasmids used this study. (b) The knockdown efficiency of UPR genes. (c) Cell proliferation of different cell lines detected by CCK8. ns, not significant. *P* values were calculated using a two-tailed *t* test, see also Supplementary File 1. Data come from three independent biological replicates. (d) Schematic overview of the tail vein injection experiment. (e) BLI imaging of distant organs from metastatic cell lines Ts15-Met with or without UPR genes KD.

The rainbow gradient bar represents the photon flux. (f) Summary of metastasis from Ts15-Met. (g) BLI imaging of distant organs from metastatic cell lines Ts8+15-Met with or without UPR genes KD. The rainbow gradient bar represents the photon flux. (h) Summary of metastasis from Ts8+15-Met.

REFERENCE

1. Gilbert, L. A. et al. Genome-scale CRISPR-mediated control of gene repression and activation. *Cell* (2014) 159: 647-661.

4) In Fig6j, proper vehicle control should be done at the same time.

Response: We thank the reviewer for pointing this out. We have updated and added these results to Supplementary Fig. 12 in the revised manuscript (here shown as **Response figure 5**).

Response figure 5. Oleuropein treatment effectively reduced aneuploid teratoma metastasis.

(a, b) BLI imaging of organs derived from trisomic teratoma-bearing mice treated with Oleuropein (a) or vehicle (b). The rainbow gradient bar represents the photon flux. Number of mice treated with oleuropein (Ts8, n=10; Ts11, n=8; Ts8+15, n=6). The bright-field image of gross lung metastasis was taken from the mouse that was dead just before BLI imaging in Ts8-vehicle group. Number of mice treated with saline (Ts8, n=6; Ts11, n=8; Ts8+15, n=7).

5) Number of literatures have demonstrated importance of UPR response in immune landscape of metastatic lesion. It might be interesting to test if this is the case for the aneuploid ES cells as well.

Response: We thank the reviewer for the comment. It was reported that activation of UPR in

cancers can initiate transcriptional programs that can shape a tumorigenic proinflammatory milieu (*Grootjans et al.*, 2016). UPR activation in prostate cancer increased the expression of proinflammatory mediators IL6 and TNF α , which might promote inflammation-induced malignancy (*Mahadevan et al.*, 2010). Recently, a study reported that UPR mechanistically linked aneuploidy and local immune dysregulation, as aneuploidy correlated with tumor disease stages while the PERK branch of UPR displayed the strongest association with a decreased cytotoxicity (CYT) score (*Xian et al.*, 2021).

In our single cell RNA-seq data, we captured 31,675 immune cells, which can be divided into granulocytes, macrophages, dendritic cells, and innate-like lymphocytes (ILCs). Copy number variation analysis showed that these cells were all diploid cells of host origin. We failed to capture T and B cells since we used the SCID-Beige mice that lack the functional B and T lymphocytes. Macrophages and neutrophils were the two most abundant immune cell populations. Macrophages were detected in all samples, while granulocytes were predominantly detected in the aneuploid samples. These infiltrating granulocytes highly expressed *Entpd1* (encoding CD39), an ectonucleoside triphosphate diphosphohydrolase which can inhibit the activation of immune cells by degrading extracellular ATP (eATP) and ADP released by damaged cells (*Li et al.*, 2019). CD39 is becoming a promising target in tumor immune therapy (*Moesta et al.*, 2020). Cell-cell communication analysis indicated that granulocytes were recruited by ES_Ori and ES_Stem through Mif signaling (*Kang et al.*, 2019; *Bucala et al.*, 2007; *Noe et al.*, 2020), and it was reported that blockade of *Mif* can effectively reduce tumorigenesis in multiple genitourinary cancers through a variety of mechanisms (*Penticuff et al.*, 2019). Thus, the selective infiltration of granulocytes could be explained by the enrichment of ES_Ori and ES_Stem in metastatic tumors. Of particular interesting, cell-cell interaction indicated that ES_Stem were more likely to evade immune cell attacks than ES_Ori. In comparison between ES_Ori and ES_Stem, ES_Stem expressed less *Pvr*, *Nectin2*, *Nectin1* which makes it harder for ES_Stem to activate receptors including *CD44*, *CD226*, *CD96*, involved in T and NK cell activation (*Bottino et al.*, 2003). Downregulation of *Cdh1* and *Icam1* in ES_Stem reduces immune cell adhesion (*Edelman et al.*, 1987). Granzyme A signaling (*van Eck et al.*, 2017) promotes the lysis and oncostatin M (OSM) limits the proliferation while corresponding receptors decreased in ES_Stem (**Response figure 6**).

Response figure 6. Immune landscape of ESC-derived lesions. (a) Bar plot showing the abundance of non-ES cell. (b) Heatmap showing the expression of ligands-receptor pairs predicted by cell-cell communication analysis. (c) Heatmap showing the *Mif* signaling in non-ES cells. (d) Violin plot displaying the expression of *Mif* in non-ES cells.

REFERENCE

- Grootjans, J., Kaser, A., Kaufman, R. J. & Blumberg, R. S. The unfolded protein response in immunity and inflammation. *Nat Rev Immunol* (2016), 16: 469-484.
- Mahadevan, N. R., Fernandez, A., Rodvold, J. J., Almanza, G. & Zanetti, M. Prostate cancer cells undergoing ER stress in vitro and in vivo activate transcription of pro-inflammatory cytokines. *J Inflamm Res* (2010), 3: 99-103.
- Xian, S. et al. The unfolded protein response links tumor aneuploidy to local immune dysregulation. *EMBO Reports* (2021), 2: e52509.
- Li, X. Y. et al. Targeting CD39 in cancer reveals an extracellular ATP- and inflammasome-driven tumor immunity. *Cancer Discov* (2019), 9: 1754-1773.
- Moesta, A. K., Li, X. Y. & Smyth, M. J. Targeting CD39 in cancer. *Nat Rev Immunol* (2020), 20: 739-755.
- Kang, I. & Bucala, R. The immunobiology of MIF: function, genetics and prospects for precision medicine. *Nat Rev Rheumatol* (2019), 15: 427-437.
- Bucala, R. & Donnelly, S. C. Macrophage migration inhibitory factor: A probable link between inflammation and cancer. *Immunity* (2007), 26: 281-285.
- Noe, J. T. & Mitchell, R. A. MIF-dependent control of tumor immunity. *Front Immunol* (2020), 11: 609948.
- Penticuff, J. C., Woolbright, B. L., Sielecki, T. M., Weir, S. J. & Taylor, J. A., 3rd. MIF family proteins in genitourinary cancer: tumorigenic roles and therapeutic potential. *Nat Rev Urol* (2019), 16: 318-328.
- Bottino, C. et al. Identification of PVR (CD155) and Nectin-2 (CD112) as cell surface ligands for the human DNAM-1 (CD226) activating molecule. *J Exp Med* (2003), 198: 557-567.

11. Edelman, G. M. CAMs and Igs: cell adhesion and the evolutionary origins of immunity. *Immunol Rev* (1987), 100: 11-45.
12. van Eck, J. A., Shan, L., Meeldijk, J., Hack, C. E. & Bovenschen, N. A novel proinflammatory role for granzyme A. *Cell Death Dis* (2017), 8: e2630.

6) Finally, it is critical to distinguish between aneuploidy and chromosomal instability. While the authors make the claim that it is aneuploidy itself that drives metastasis there is prior work suggesting that the ongoing instability can also drive metastasis. While the two are related and linked they are not synonymous. The authors should address this point experimentally.

Response: We thank the reviewer for the comments. Aneuploidy is considered as the outcome of chromosomal instability (CIN), and often coexists with CIN in cancer. Prior work reported that CIN can drive metastasis through a cytosolic DNA response (*Bakhoun et al.*, 2018). In this study, we aimed to investigate the role of aneuploidy in teratoma metastasis. Whole genome sequencing (WGS) of primary teratomas and paired metastases showed that most metastases preserved CNV patterns similar to those of primary teratomas, indicating that aneuploid cells are relatively stable during metastasis.

To test the mitotic state of aneuploid mESCs, we labelled WT and aneuploid ESCs with H2B-mCherry, and visualized cell division using time-lapse imaging. We found the mitotic time (from envelope breakdown to anaphase onset) were similar between WT and aneuploid mESCs. Moreover, the mitotic error rates in aneuploid mESCs are no more than 7%, which is comparable with those in WT mESCs. Thus, aneuploid ESCs exhibited equivalent CIN levels with WT mESCs (**Response figure 7**). Besides, we have updated and added these results to Supplementary Fig. 8 and Supplementary videos 1-3 in the revised manuscript.

Response figure 7. Comparable CIN levels in WT and aneuploid mouse ESC lines during mitosis. (a) Statistical analysis of the mitotic time (from envelope breakdown to anaphase onset) in WT and aneuploidies. WT, n=96; n=100 (Ts6, Ts11, Ts8+15); Ts8, n=105. (b) Quantification of mitotic error rates in WT and aneuploid ESC lines. For each group, n >100.

REFERENCE

1. Bakhoun, S. F. et al. Chromosomal instability drives metastasis through a cytosolic DNA response. *Nature* (2018), 553: 467-472.

All the major changes in the revised manuscript are marked in red.

Response to Reviewer #2

Reviewer #2 (Remarks to the Author):

Aneuploid embryonic stem cells drive teratoma metastasis

“Aneuploid embryonic stem cells drive teratoma metastasis” manuscript shows how several chromosome gains (trisomies), in the context of stable CIN, promote teratoma invasiveness and metastasis. This is a novel and interesting finding, in the current discussion whether aneuploidy can affect tumor progression by itself without requiring/inducing chromosome instability. However, the authors should address major and minor comments prior acceptance of the manuscript:

Major comments:

1- Seems that a lot of effect is driven by the gain of chromosome 8.

- a. Figure 1: The % of metastasis developed ranges from 31% to 63% depending of the background, but the highest rates are consistently found in the context of chromosome 8 trisomy (Ts8 (62.5%) Ts6+8 (45%) Ts8+15 (62.9%)).

Response: We thank the reviewer for the comment. Gain of chromosome 8 in mouse ESCs is the most frequently observed chromosome event during prolonged cell culture, which confers a proliferation benefit to ESCs (Liu *et al.*, 1997; Santaguida *et al.*, 2015). In our study, single trisomic mESC lines were obtained by genetic manipulation and double drug selection (Zhang *et al.*, 2016). Unlike single trisomies, double trisomic mESC lines were established from single trisomies followed by the spontaneous chromosomes gains, which frequently resulted in double trisomies harboring an extra chromosome 8 (*e.g.* Ts6+8 and Ts8+15 mESCs). In fact, 50% of aneuploid cell lines used in this study harboring an extra chromosome 8, which might create the illusion that lots of effects are associated with gain of chromosome 8.

In fact, the percentage of metastasis varies among different trisomies. For 4 single-trisomic ESC lines, the metastasis rates are 31.25%, 62.5%, 31.58% and 44.44% for Ts6, Ts8, Ts11 and Ts15 respectively. Anyhow, all trisomic but not WT mESCs possess metastatic capability. Teratoma metastasis is driven by the aneuploidy event.

REFERENCE

1. Liu X, Wu H, Loring J, Hormuzdi S, Disteche CM, Bornstein P, Jaenisch R. Trisomy 8 in ES cells is a common potential problem in gene targeting and interferes with germ line transmission. *Dev Dyn* (1997), 209: 85-91.
 2. Santaguida, S. & Amon, A. Short- and long-term effects of chromosome mis-segregation and aneuploidy. *Nat Rev Mol Cell Bio* (2015), 16: 473-485.
 3. Zhang, M. L. et al. Aneuploid embryonic stem cells exhibit impaired differentiation and increased neoplastic potential. *EMBO J* (2016) 35: 2285-2300.
-
- b. Figure 2 & Extended Figure 4: WGS revealed one Ts6 changes to Ts6+8 and one Ts8 changes to Ts6+8

Response: As shown in Figure 2a, WGS of primary teratomas and metastatic lesions showed

that most metastases presented CNV patterns similar to those of primary teratomas. Extra chromosome gains occasionally occurred in few metastatic lesions, such as gains of chromosome 8 in Ts6-4-M and gains of chromosome 6 in Ts8-2-M (Supplementary Fig. 4b), which were not the common events during teratoma metastasis.

c. Figure 4: CTCs were only successfully isolated from Ts6+8

Response: In our study, the CTC-like cells were isolated from teratoma-bearing mice before the teratomas reached 1.5 cm in diameter and did not undergo teratoma surgery. Generally, the CTCs are extremely rare in peripheral blood. Of them, only a small part can be captured from the peripheral blood mononuclear cells. Successful culture of CTCs is even harder since the appropriate culture condition is unknown. We are very lucky to successfully isolate and culture CTCs from two mice bearing Ts6+8 teratomas under the culture conditions for ESCs. The isolation of CTCs from mice bearing other trisomic teratomas is still ongoing.

d. Figure 6a: Proteasome activity reduction significant only in Ts6 and Ts8

Response: Proteasome activity in ESCs is expected to elevate in the early stage (day 2) of EB differentiation to degrade damaged proteins, including carbonylated and advanced glycation end proteins. Then it declines in the late stage of EB differentiation (*Hernebring et al., 2006*). Like that, WT cells exhibited elevated proteasome activity at day 2 of EB differentiation and decreased proteasome activity at day 6 of EB differentiation (the original Figure 6a, now Supplementary Fig. 11 in the revised manuscript). However, the proteasome activity in all aneuploid cells (not only in Ts6 and Ts8) was markedly decreased at day 2 of EB differentiation, which is in marked contrast to WT cells.

REFERENCE

1. Hernebring, M., Brolen, G., Aguilaniu, H., Semb, H. & Nystrom, T. Elimination of damaged proteins during differentiation of embryonic stem cells. *Proc Natl Acad Sci USA* (2006), 103: 7700-7705.

e. Mice Chromosome 8 contains 75% of the genes encoding for rRNAs. (<https://www.ncbi.nlm.nih.gov/genome?term=txid10091%5BOrganism%3Aanoexp%5D&cmd=DetailsSearch>) This might be also affecting the results of the pathways affected in extended figure 7d, since samples carrying trisomy 8 are one of the main contributors to this analysis (ES_Stem vs ES_Ori)

All this data should be mentioned in the manuscript and included in the discussion with rationale about possible explanations for this phenomenon.

Response: To test whether trisomy 8 affects the results of the pathways in the original extended figure 7d (the Supplementary Fig. 10d in the revised manuscript), we excluded all samples with extra chromosome 8 and performed pathway enrichment analysis on differentially expressed genes (DEGs) between ES_Stem and ES_Ori. Among the upregulated terms in this analysis, “negative regulation of cell population proliferation” and “positive regulation of cell death” are

listed as the top 2 enriched terms in the original analysis. “Translation”, “metabolism of RNA”, “cell cycle” and “ribonucleoprotein complex biogenesis” are the top 4 enriched terms of the downregulated DEGs in the original analysis as well, regardless of with or without analysis of samples carrying trisomy 8 (**Response figure 8**).

Response figure 8. Pathway enrichment analysis on DEGs between ES_Stem and ES_Ori when excluding cells originated from Ts8 cell line.

2- ScRNAseq comments:

a. From figure 5c and extended figure 7c, it seems that the contribution to the ES_Stem population is highly enriched with cells from Ts11_M. In the data from InferCNV, gain of chr11 is the only clearly detectable trisomy traced in this cell population and it's present in almost 50% of the cases. Also the ES_Stem population existing in the WT samples represents only 1% of the total cells on the samples. So the comparison in figure 5f comparing differentially expressed genes between ES_Ori or ES_Stem between WT and AC might be biased by the enrichment of specific genotypes.

Response: We thank the reviewer for the comment. We excluded samples with trisomic chromosome 11 and performed Pathway enrichment analysis on the remaining samples. We still observed significant downregulation of genes related to proteasomes in ES_Stem and enrichment of related genes in the "RNA metabolism" pathway (**Response figure 9**). In addition, we compared the different aneuploid cell groups separately. To ensure the representativeness of the results, we only compared subpopulations with more than 100 cells. The results showed that compared with ES_Ori, genes related to proteasomes were down-regulated in all groups of ES_Stem (**Response figure 10**). Therefore, the downregulation of genes related to proteasomes in ES_Stem was not caused by a specific group of aneuploid cells but a cross-group phenomenon.

Response figure 9. Pathway enrichment analysis on DEGs between ES_Stem and ES_Ori when excluding cells originated from Ts11 cell line.

Response figure 10. Transcriptional changes between ES_stem and ES_Ori. Volcano plot showing DEGs between ES_stem of Ts6 and ES_Ori of Ts6 (a), between ES_stem of Ts6+8 and ES_Ori of WT (b), between ES_stem of Ts6 and ES_Ori of WT (c), between ES_stem of Ts8 and ES_Ori of Ts8 (d), between ES_stem of Ts8 and ES_Ori of WT (e), between ES_stem of Ts8+15 and ES_Ori of WT (f), between ES_stem of Ts11 and ES_Ori of Ts11 (g), between ES_stem of Ts11 and ES_Ori of WT (h).

b. Extended figure 7b seems to show that metastatic samples are enriched in cells from the immune compartment coming from the mice (DCs and Macrophages). This might be worth some discussion.

Response: We thank the reviewer for the suggestions. We captured 31,675 immune cells, which

can be divided into granulocytes, macrophages, dendritic cells, and innate-like lymphocytes (ILCs). Copy number variation analysis showed that these cells were all diploid cells of host origin. We failed to capture T and B cells since we used the nude mouse. Macrophages and neutrophils were the two most abundant immune cell populations. Macrophages were detected in all samples, while granulocytes were predominantly detected in the aneuploid samples. These infiltrating granulocytes highly expressed *Entpd1*(encoding CD39), an ectonucleoside triphosphate diphosphohydrolase which can inhibit the activation of immune cells by degrading extracellular ATP (eATP) and ADP released by damaged cells (*Li et al.*, 2019). CD39 is becoming a promising target in tumor immune therapy (*Moesta et al.*, 2020). Cell-cell communication analysis indicated that granulocytes were recruited by ES_Ori and ES_Stem through Mif signaling (*Kang et al.*, 2019; *Bucala et al.*, 2007; *Noe et al.*, 2020), and it was reported that blockade of *Mif* can effectively reduce tumorigenesis in multiple genitourinary cancers through a variety of mechanisms (*Penticuff et al.*, 2019). Thus, the selective infiltration of granulocytes could be explained by the enrichment of ES_Ori and ES_Stem in metastatic tumors. Of particular interesting, cell-cell interaction indicated that ES_Stem are more likely to evade immune cell attacks than ES_Ori. In comparison between ES_Ori and ES_Stem, ES_Stem expressed less *Pvr*, *Nectin2*, *Nectin1* which makes it harder for ES_Stem to activate receptors including *CD44*, *CD226*, *CD96*, involved in T and NK cell activation (*Bottino et al.*, 2003). Downregulation of *Cdh1* and *Icam1* in ES_Stem reduces immune cell adhesion (*Edelman et al.*, 1987). Granzyme A signaling (*van Eck et al.*, 2017) promotes the lysis and oncostatin M (OSM) limits the proliferation while corresponding receptors decreased in ES_Stem (**Response figure 11**).

Response figure 11. Immune landscape of ES lesion. (a) Bar plot showing the abundance of non-ES cell. (b) Heatmap showing the expression of ligands-receptor pairs predicted by cell-cell communication analysis. (c) Heatmap showing the *Mif* signaling in non-ES cells. (d) Violin plot displaying the expression of *Mif* in non-ES cells.

REFERENCE

- Li, X. Y. et al. Targeting CD39 in cancer reveals an extracellular ATP- and inflammasome-driven tumor immunity. *Cancer Discov* (2019), 9: 1754-1773.
- Moesta, A. K., Li, X. Y. & Smyth, M. J. Targeting CD39 in cancer. *Nat Rev Immunol* (2020),

20: 739-755.

3. Kang, I. & Bucala, R. The immunobiology of MIF: function, genetics and prospects for precision medicine. *Nat Rev Rheumatol* (2019), 15: 427-437.
4. Bucala, R. & Donnelly, S. C. Macrophage migration inhibitory factor: A probable link between inflammation and cancer. *Immunity* (2007), 26: 281-285.
5. Noe, J. T. & Mitchell, R. A. MIF-dependent control of tumor immunity. *Front Immunol* (2020), 11: 609948.
6. Penticuff, J. C., Woolbright, B. L., Sielecki, T. M., Weir, S. J. & Taylor, J. A., 3rd. MIF family proteins in genitourinary cancer: tumorigenic roles and therapeutic potential. *Nat Rev Urol* (2019), 16: 318-328.
7. Bottino, C. et al. Identification of PVR (CD155) and Nectin-2 (CD112) as cell surface ligands for the human DNAM-1 (CD226) activating molecule. *J Exp Med* (2003), 198: 557-567.
8. Edelman, G. M. CAMs and Igs: cell adhesion and the evolutionary origins of immunity. *Immunol Rev* (1987), 100: 11-45.
9. van Eck, J. A., Shan, L., Meeldijk, J., Hack, C. E. & Bovenschen, N. A novel proinflammatory role for granzyme A. *Cell Death Dis* (2017), 8: e2630.

c. Figure 5e: The authors in line 213 quote “we noted that cancer-related genes such as S100 family members (S100a8, S100a9, S100g)^{28,29} and Hbb-bs³⁰ were upregulated”. According to the figure these genes are apparently upregulated in intermediate timepoints but not at the latest timepoint. Additionally, there is no information provided about the other clusters in the figure, so the authors should describe these clusters or remove them from the figure.

Response: We thank the reviewer for pointing this out. We have removed the description and annotations of clusters of S100 family members (S100a8, S100a9, S100g) and Hbb-bs in the revised Figure 5e and 5f.

d. The authors do not indicate whether the primary and metastases samples are paired. And the data from extended table 1 and 2 does not clarify this fact. The Supplementary tables 1 and 2 should be indicate which samples are primary teratomas and which ones are metastasis.

Response: We thank the reviewer for pointing this out. In fact, the samples used for WGS/WES were paired primary teratomas and metastases, while samples that were used for single-cell RNA sequencing were not paired. We have added the metastases information in the revised Supplementary Tables 1 and 2. The detail information was shown in NCBI accompanied with sequencing raw data (PRJNA790979).

3- Proteasome activity and differentiation:

Response: To address the key point of “Aneuploidy-induced metastasis can be inhibited by proteasome activator or UPR inhibitor”, we further explored the effects of UPR inhibition by generating UPR genes (*Atf6*, *Eif2a* and *Xbp1*) knockdown metastatic cells of Ts15-Met and Ts8+15-Met via CRISPR interference (the new Supplementary Fig. 14 in the revised manuscript). Metastasis was routinely monitored by BLI after tail vein injection of Ts15/Ts8+15-Met cells (the

new Fig. 6e). We found Ts15-Met and Ts8+15-Met cells with UPR gene knockdown exhibited decreased metastasis efficiency compared with control (the new Fig. 6f-6i). Considering these new animal studies are very solid to support the key point and there are obvious limitations of EB differentiation in studying metastasis, we rearranged the panels in **Figure 6** and **moved the results of EB differentiation to the Supplementary Fig. 11**.

a. In line 239 authors quote “Unlike WT cells, the total proteasome activity in aneuploid cells quickly declined during the early stage of embryoid body (EB) formation (Fig. 6a).”. However, in figure 6a at the D6 timepoint, the WT proteasome activity has declined at a similar level of 50% of the aneuploid samples. This might be related to specific trisomies rather than a general aneuploidy/trisomy effect.

Response: We apologize for not clearly defining “the early stage of EB formation” in the original manuscript. The early stage of EB formation means day 2 of EB differentiation. To avoid misunderstanding, we changed “the early stage of EB formation” to “day 2 of EB formation” in the revised manuscript. WT cells exhibit elevated proteasome activity at day 2 of EB differentiation and decreased proteasome activity at day 6 of EB differentiation (the original Figure 6a, now Supplementary Fig. 11 in the revised manuscript). However, the proteasome activity in all aneuploid cells (not only in Ts6 and Ts8) was decreased at day 2 of EB differentiation, which was in marked contrast to WT cells.

b. I would like to see for figures 6d and 6f a foldchange graph between timepoint D0 and D6 or D2 and D4 if there is only data available from these last two timepoints.

Response: We thank the reviewer for pointing this out. The fold change graph for the original Figures 6d and 6f (now Supplementary Fig. 11e, 11g in the revised manuscript) is shown below (**Response figure 12**).

Response figure 12. Fold change of the expression of some proteasome subunits (PA28α, β5i) and ER stress proteins (BIP, p-PERK, p-EIF2A, ATF6) between timepoints D2 and D4 (relative to WT).

c. In line 254 quote “Oleuropein effectively reduced aneuploid teratoma volume and partially rescued the deficiency of aneuploid ESC differentiation, as evidenced by significantly decreased proportions of Oct4+ cells in the teratoma”. Figure 6h actually shows WT oct4 expression under treatment but not vehicle. It seems that the Oleuropein effect is more related to contain the OCT4+ cells in dense clusters hampering their spread, but they remain highly positive for OCT4. Maybe this is affecting the invasiveness potential of OCT4+ cells. I suggest testing in vitro invasion,

migration, and proliferation under treatment.

Response: We thank the reviewer for the comment. Actually, a small number of OCT4⁺ cells existed not only in Oleuropein-treated but also in vehicle WT teratomas (**Response figure 13**). After Oleuropein treatment, the number of OCT4 positive regions reduced significantly in aneuploid teratomas, while the treatment had negligible effect on WT teratomas. The reviewer is concerned about whether Oleuropein is more related to contain the OCT4⁺ cells in dense clusters, we therefore purified the population of OCT4⁺ cells from the primary cultured teratoma cells and tested their migration abilities under the treatment of Oleuropein by transwell assay. We found Oleuropein did not affect OCT4⁺ teratoma cell migration except for Ts11 (**Response figure 14**). We also tested the proliferation abilities of the primary teratoma cells and found Oleuropein treatment had negligible effect on teratoma cell proliferation (**Response figure 15**).

Response figure 13. Representative immunohistochemistry staining (OCT4) images of WT teratomas (up: vehicle; below: +Oleuropein). Scale bars, 100 μ m.

Response figure 14. The migration abilities of OCT4⁺ teratoma cells treated with Oleuropein. * $p = 0.0037$, $n=10$. P values were calculated using a two-tailed t test. Data come from three independent biological replicates.

Response figure 15. Oleuropein treatment has no obvious effect on the proliferation of teratoma cells. For each group, $n=3$.

d. Additionally, the authors do not provide in vitro evidence showing that Oleuropein rescues proteasome activity in the ESC cell lines used prior to in vivo treatment.

Response: We thank the reviewer for pointing this out. In fact, we had tested the proteasome activity of cultured aneuploid cells with or without Oleuropein treatment during the EB differentiation. We found that Oleuropein treatment could increase the proteasome activity in aneuploid cells, especially in Ts6, Ts8 and Ts11 (**Response figure 16a**). Moreover, Oleuropein promoted aneuploid EB formation with large cystic structures (**Response figure 16b**). We have updated and added these results to the Supplementary Fig. 11m-n in the revised manuscript.

Response figure 16. Oleuropein rescued proteasome activity of aneuploid EBs and promote EB differentiation. (a) Proteasome activity during EBs formation on D0~D4. Error bars, \pm SD. *P* values were calculated using a two-tailed *t* test, see also Supplementary File 1. Data come from three independent biological replicates. (b) *In vitro* EB differentiation of WT and aneuploid ESCs with or without Oleuropein treatment. Black arrows point to the representative large cystic structures of differentiated EBs. Scale bars, 200 μ m.

e. In line 252 quote “These results suggest that insufficiency of proteasome activity and overactivated UPR might underlie the differentiation defects of aneuploid cells.” I don’t think the authors actually showed data regarding differentiation, but rather stemness maintenance. To support this would be great to have Histology, IHC, gene expression data and/or protein levels on WT and AC teratoma samples.

Response: We thank the reviewer for the comment. From the scRNA-seq analysis, we found that no transient upregulation of proteasome subunit genes was observed along the pseudotime trajectory during aneuploid ESC differentiation (Fig.5e), therefore we used embryoid bodies (EBs) formation to test the proteasome activity during different stages of ESC differentiation. We found the total proteasome activity was decreased and many carbonyl proteins were accumulated in aneuploid cells in the early stage of differentiation (D2 of EB formation), which further triggered ER stress and UPR, underlying the stemness maintenance in aneuploid ESC-derived EBs and teratomas.

In addition to the retention of OCT4⁺ cells, aneuploid ESCs indeed exhibited differentiation defects. In our previous work (Zhang *et al.*, 2016), we have shown that aneuploid ESC-derived EBs were more compact and smaller at day 8 of differentiation compared with the cavitated and large cystic EB structures formed by WT ESCs. RT-qPCR showed that aneuploid EBs aberrantly expressed the ectoderm marker genes (*Fgf5*, *Mash1*), mesoderm marker genes (*Brachyury*, *Bmp4*, *Hand1*) and endoderm marker genes (*Foxa2*, *Gata6*) (**Response figure 17**). Moreover, large proportions of more primitive undifferentiated or low differentiated regions were detected in the trisomic ESC-derived teratomas (**Response figure 18**). These results indicated that the differentiation timing of trisomic ESCs to multiple lineages had been delayed in the aneuploid cells.

Response figure 17. Limited differentiation abilities of aneuploid ESCs *in vitro*. Time course analysis of marker gene expression during EB formation by RT-qPCR. Error bars, SD. *n* = 3. (Cited from Zhang *et al.*, *EMBO J*, 2016; Fig. 3E).

Response figure 18. Representative images of undifferentiated regions in Ts8 ESC-derived teratomas (middle, top, and bottom) and the primitive neural rosettes in the teratomas derived from Ts15 ESCs (right, top, and bottom). The images of mature neuroectoderm from wild-type ESC-formed teratomas (left, top, and bottom) are shown. The bottom panels are the enlarged views of the white dotted boxes of the top panels. Hematoxylin and eosin staining. Scale bars, 50 μ m. (Cited from *Zhang et al., EMBO J*, 2016; Fig. 4C).

REFERENCE

1. Zhang, M. L. et al. Aneuploid embryonic stem cells exhibit impaired differentiation and increased neoplastic potential. *EMBO J* (2016) 35: 2285-2300.

f. Bulk RNAseq in differentiated EBs from WT or trisomic cell will help to elucidate the effect of these gains in differentiation and tumor invasion and metastasis.

Response: We appreciate the helpful suggestions from the reviewer. We have collected EBs from different aneuploid ESC lines and WT ESCs at day 8 of differentiation, and performed bulk RNA-seq. Generally consistent with the scRNA-seq data, aneuploid EBs showed increased stemness score and decreased endoderm and mesoderm development scores (**Response figure 19a-19d**). Besides, ER stress (*Chen et al., 2021*) score of aneuploid EBs was higher than WT as well (**Response figure 19e**). Further pathway enrichment analysis showed that extracellular matrix organization was significantly down-regulated in aneuploid EB cells, which might be associated with increased tumor invasion and metastasis (*Pickup et al., 2014; Gilkes et al., 2014*) (**Response figure 19f**). We have updated and added these results to Supplementary Fig. 11 in the revised manuscript.

Response figure 19. Aneuploid EBs displayed enhanced stem cell traits, reduced differentiation capacity and increased ER stress. Violin plots showed the scores of stemness (a), endoderm development (b), mesoderm development (c), ectoderm development (d), and ER stress (e) gene sets in differentiated EBs (day 8, D8). The stemness gene set consists of *Pou5f1*, *Nanog*, *Zfp42*, and *Dppa5a*. *P* values were calculated using a two-tailed *t* test. (f) Pathway enrichment of shared DEGs in the comparison between aneuploid and WT EBs. Data come from three independent biological replicates.

REFERENCE

- Chen, X. & Cubillos-Ruiz, J. R. Endoplasmic reticulum stress signals in the tumour and its microenvironment. *Nat Rev Cancer* (2021), 21: 71-88.
- Pickup, M. W., Mouw, J. K. & Weaver, V. M. The extracellular matrix modulates the hallmarks of cancer. *EMBO Rep* (2014), 15: 1243-1253.
- Gilkes, D. M., Semenza, G. L. & Wirtz, D. Hypoxia and the extracellular matrix: drivers of tumour metastasis. *Nat Rev Cancer* (2014), 14: 430-439.

Minor Comments

4- CTCs experiment

a. In my opinion, the most interesting question in this experiment is missing. Finding CTCs when there is already metastasis present it is expected. Would have been more interesting to withdraw blood before removing the primary teratoma to avoid any possible bias introduced during the surgery.

Response: We thank the reviewer for giving us the opportunity to clarify the experimental procedure. In fact, to avoid the potential contamination from the surgery, we isolated CTC-like cell lines (both #1 and #2) from Ts6+8 teratoma bearing mice that did not undergo teratoma

surgery. We have added the description in the revised manuscript (Methods: page 18, lines 383-384).

b. Extended figure 6b. The authors showed positivity for OCT4 but they do not show that the cells retained in vitro eGFP.

Response: We thank the reviewer for pointing it out. Ts6+8 CTC-like cells are positive for eGFP (**Response figure 20**). The results were supplemented as new Supplementary Figure 9b in the revised manuscript.

Response figure 20. Bright-field and eGFP images of Ts6+8 CTC-like cells.

c. The authors should address any possible limitations of the CTCs study in the discussion including why CTCs were only successfully isolated from Ts6+8.

Response: The related discussion on the possible limitations of CTCs study has been supplemented in the revised main text (Page 14, lines 273-283).

5- Trisomy correction: the image showing the phenotype change from Ts11 to Di11 (figure 3b) doesn't look really different. Authors should provide some quantitative assessment/measurement to prove this or select a better image.

Response: We have quantified the size of EBs and added more pictures. These results were supplemented as new Supplementary Figure 6 in the revised manuscript (here shown as **Response figure 21**).

Response figure 21. Quantification of EB diameters.

(a) Morphologies of EB formation. Images show EBs from WT, Ts11, Ts8 and the isogenic diploid mESCs (named Di8 and Di11, respectively). Scale bars, 200 µm. (b) Quantification of EB diameters (WT, Di8, n=5; Di11, n=7; Ts8, Ts11, n=15). Ts, trisomy. Di, diploid. Error bars, ± SD. ns, not significant. *P* values were calculated using a two-tailed *t* test.

6- The manuscript talks about aneuploidy in general when it is only focused on gain of specific chromosomes (Trisomies). This is relevant because aneuploidy also includes chromosome deletion. I would suggest to change the manuscript title from Aneuploidy to Trisomy.

Response: We thank the reviewer for the suggestion. Because mammalian cells and organisms harboring somatic chromosome loss usually cannot survive, our study used trisomies to model the unbalanced chromosome events. In fact, a variety of studies on trisomies used aneuploidy as the keywords of the title (*Williams et al.*, 2008; *Tang et al.*, 2011; *Stamoulis et al.*, 2019; *Replogle et al.*, 2020; *Hwang et al.*, 2021). So, it would be better to keep “Aneuploidy” in the title.

REFERENCE

1. Williams, B. R. *et al.* **Aneuploidy** affects proliferation and spontaneous immortalization in mammalian cells. *Science* (2008), 322: 703-709.
2. Tang, Y. C., Williams, B. R., Siegel, J. J. & Amon, A. Identification of **aneuploidy**-selective antiproliferation compounds. *Cell* (2011), 144: 499-512.
3. Stamoulis, G. *et al.* Single cell transcriptome in **aneuploidies** reveals mechanisms of gene dosage imbalance. *Nat Commun* (2019), 10: 4495.
4. Replogle, J. M. *et al.* **Aneuploidy** increases resistance to chemotherapeutics by antagonizing

cell division. *Proc Natl Acad Sci USA* (2020), 117: 30566-30576.

5. Hwang, S. *et al.* Consequences of **aneuploidy** in human fibroblasts with trisomy 21. *Proc Natl Acad Sci USA* (2021), 118: e2014723118.

7- The discussion section need to be more elaborated, commenting on possible caveats and limitations of the presented study and comparing the results to previous findings from other groups.

Response: We thank the reviewer for the suggestion. The discussion section has been expanded and elaborated in the revised manuscript (Pages 13-14, lines 249-283).

All the major changes in the revised manuscript are marked in red.

REVIEWER COMMENTS

Reviewer #1 (Remarks to the Author):

The authors have done a good job addressing all my comments. The only remaining one is ruling in or out CIN vs. aneuploidy. They perform some analysis of mitotic errors which reveals elevated mitotic errors in the aneuploid cells vs. WT controls. It is unclear how they performed these analyses, what images look like and whether H2B method is sufficiently sensitive to conduct this analysis. I would suggest simply doing a fixed cell based IF microscopy analysis of anaphase chromosome segregation and to score lagging chromosomes/acentric chromatic fragments and chromatin bridges as a function of anaphase (the fraction of anaphase cells with errors divided by the total anaphase cells). I would not merge micronuclei in this analysis. Micronuclei should be separately reported as fraction of primary nuclei (micronucle/primary nuclei). Finally this analysis should be statistically powered to test for significance. Otherwise this is a great contribution and I am fully supportive of publication once this issue has been addressed.

Reviewer #2 (Remarks to the Author):

The authors did a great job responding to original comments. I just have a few follow-up comments:

1. Oleuropenin reduces teratoma volume of aneuploid clones to values more similar to teratoma volume in wildtype. Could the volume reduction be the cause of the metastasis? I think the language needs to be softened to reflect this possibility.
2. In response to initial comments, the authors included information from their previous publication related to EB differentiation. Although already published, if the authors could add some of this explanation in the results section, it would be quite helpful.

Point-by-point response

Manuscript ID: NCOMMS-23-04507-A

We are pleased to learn that the reviewers are satisfied with the changes introduced in the revised manuscript and with our previous responses. Once again, we sincerely thank the reviewers for their appreciation of our work and for their valuable comments which helped us improve the quality of the manuscript. Please see our point-by-point responses below.

Response to Reviewer #1

Reviewer #1 (Remarks to the Author):

The authors have done a good job addressing all my comments. The only remaining one is ruling in or out CIN vs. aneuploidy. They perform some analysis of mitotic errors which reveals elevated mitotic errors in the aneuploid cells vs. WT controls. It is unclear how they performed these analyses, what images look like and whether H2B method is sufficiently sensitive to conduct this analysis. I would suggest simply doing a fixed cell based IF microscopy analysis of anaphase chromosome segregation and to score lagging chromosomes/acentric chromatic fragments and chromatin bridges as a function of anaphase (the fraction of anaphase cells with errors divided by the total anaphase cells). I would not merge micronuclei in this analysis. Micronuclei should be separately reported as fraction of primary nuclei (micronucle/primary nuclei). Finally this analysis should be statistically powered to test for significance. Otherwise this is a great contribution and I am fully supportive of publication once this issue has been addressed.

Response: We thank the reviewer for the comment. In our study, we detected the CIN status of WT and aneuploid ES cells by time-lapse imaging using a histone 2B-mcherry (H2B-mCherry) reporter. Transfected cells with H2B-mCherry reporter is one of the well-established and minimally invasive approaches for visualization of chromatin in live cells, and very useful for real-time imaging to observe mitotic cell division (<https://www.thermofisher.cn/cn/zh/home/references/molecular-probes-the-handbook/probes-for-organelles/nuclear-and-chromosome-counterstaining-and-nissl-stains.html>).

Then time-lapse imaging of cells labeled with H2B-mcherry was performed by using Cell Discover 7 (CD7, Zeiss). Images were captured each 2 min (WT, Ts6, Ts8, Ts11) or 3 min (Ts8+15) lasted for 20-24 hours and 9 scenes were set. Images of mCherry positive cells were captured using CD7 (Zeiss) at 100× magnification. Then we analyzed the mis-segregation events like chromosome bridge, lagging chromosome, micronuclei, and multiple division of each cell line to count mis-segregation rate between aneuploid mouse ESC lines and WT (euploid mouse ESC line).

We re-analyzed chromosome mis-segregation events from two aspects: time required for mitosis from nuclear envelope breakdown to anaphase onset, and the frequency of mis-segregation events (three replicates were used for the analysis). We found that both trisomic and WT ESCs displayed relatively equal CIN levels during mitosis (**Response figure 1 and Response figure 2**), indicating trisomic cells maintained their chromosome number and propagated in a stable manner generally. Meanwhile, the fraction of primary nuclei (micronuclei/primary nuclei) was calculated as well

(Response figure 3), we have added these results to the revised manuscript (Supplementary Fig. 8, Supplementary File 1 and Supplementary videos 1-3).

Response figure 1. Representative time-lapse images showing normal mitosis and mis-segregation of WT and aneuploid ES cells labeled with H2B-mcherry. Continuous images were captured and the time was marked as magenta. Mis-segregation events were pointed by yellow arrows.

Response figure 2. Comparable CIN levels in WT and aneuploid mouse ESC lines during mitosis. (a) Statistical analysis of the mitotic time (from envelope breakdown to anaphase onset) in WT and aneuploidies. The data come from three independent experiments ($n > 60$). P values were calculated using a two-tailed t test, see also Supplementary File 1. (b) Quantification of mitotic error rates in WT and aneuploid ESC lines. The data come from three independent experiments ($n > 60$). P values were calculated using a two-tailed t test, see also Supplementary File 1.

Response figure 3. Micronuclei/primary nuclei in WT and aneuploid mouse ESC lines during mitosis. The micronucleus were pointed by yellow arrows. Micronuclei/primary nuclei was calculated and displayed in the right panel. P values were calculated using a two-tailed t test, see also Supplementary File 1.

All the major changes in the revised manuscript are marked in red.

Response to Reviewer #2

Reviewer #2 (Remarks to the Author):

The authors did a great job responding to original comments. I just have a few follow-up comments:

1. Oleuropein reduces teratoma volume of aneuploid clones to values more similar to teratoma volume in wildtype. Could the volume reduction be the cause of the metastasis? I think the language needs to be softened to reflect this possibility.

Response: We appreciate the reviewer's suggestions. Considering that Oleuropein can effectively suppress teratoma growth, we would like to discuss it in the manuscript (Discussion section, page 14, line 270-272).

2. In response to initial comments, the authors included information from their previous publication related to EB differentiation. Although already published, if the authors could add some of this explanation in the results section, it would be quite helpful.

Response: We thank the reviewer for the comment. We have added EB formation characteristics described in our previous publication in the revision (Please see "**Aneuploid embryoid bodies exhibited proteasome dysfunction and overactivated ER stress**", page 10, line 192-194).

All the major changes in the revised manuscript are marked in red.

REVIEWERS' COMMENTS

Reviewer #1 (Remarks to the Author):

I appreciate the authors' attempts to address my remaining comments. They have partially done this. The reason I do not think their data definitively rules out CIN (in addition to aneuploidy) is because there is a trend (and in some instances significant) increase in micronuclei e.g. Ts11 in response figure 3. Furthermore, the absolute fraction of micronuclei is ~10% (0.1 micronuclei/primary nuclei). This is a significant number of micronuclei and it is reminiscent of what is seen in chromosomally unstable cancer cells. Therefore the most likely explanation is that the live-cell imaging is not sufficiently sensitive to detect all anaphase missegregation events (e.g. small arms, fine bridges etc) and that the micronuclei frequency suggests that there may be ongoing CIN that is not adequately quantified by the methods used in this paper. My suggestion to address this would be to tone down the results related to CIN and not to claim that these teratomas are maintaining chromosomal stability (10% micronuclei is not a stable genome and is equivalent to highly chromosomally unstable cancer cells). The alternative would be to more rigorously measure CIN but I am appreciative that at this stage of the manuscript, modifying the results and discussion to not make strong statements about CIN in the face of inconclusive negative data would be an appropriate path forward.